# A model-based analysis of foliar $NO_x$ deposition

Erin R. Delaria[1], Ronald C. Cohen[1,2]

[1]Department of Chemistry, University of California Berkeley, Berkeley, CA, USA
[2]Department of Earth and Planetary Science, University of California Berkeley, Berkeley, CA, USA

*Correspondence to*: Ronald C. Cohen (rccohen@berkeley.edu)

**Abstract.**

Foliar deposition of $NO_2$ removes a large fraction of the global soil-emitted $NO_x$. Understanding the mechanisms of $NO_x$ foliar loss is important for constraining surface ozone, $NO_x$ mixing ratios, and assessing the impacts of nitrogen inputs to ecosystems. We have constructed a 1D multi-box model with representations of chemistry and vertical transport to evaluate the impact of

leaf-level processes on canopy-scale concentrations, lifetimes, and canopy fluxes of $NO_x$. Our model is able to closely replicate canopy fluxes and above-canopy $NO_x$ daytime mixing ratios observed during two field campaigns, one in a western Sierra Nevada pine forest (BEARPEX-2009) and the other a northern Michigan mixed hardwood forest (UMBS-2012). We present a conceptual argument for the importance of $NO_2$ dry deposition and demonstrate that $NO_2$ deposition can provide a mechanistic explanation for the canopy reduction of $NO_x$. We show that foliar deposition can explain observations suggesting

as much as ~60% of soil-emitted $NO_x$ is removed within forest canopies. Stomatal conductances greater than 0.1 cm s$^{-1}$ result in modelled canopy reduction factors in the range of those used in global models, reconciling inferences of canopy $NO_x$ reduction with leaf-level deposition processes. We show that incorporating parameterizations for vapor pressure deficit and soil water potential has a substantial impact on predicted $NO_2$ deposition in our model, with the percent of soil $NO_x$ removed within one canopy increasing by ~15% in wet conditions compared to dry conditions. $NO_2$ foliar deposition was also found to

have a significant impact on ozone and nitrogen budgets under both high and low $NO_x$ conditions.

## 1 Introduction

The chemistry of nitrogen oxides ($NO_x \equiv NO + NO_2$) has a large impact on the oxidative capacity of the atmosphere and the budget of global surface ozone (Crutzen, 1979). $NO_x$ is primarily removed from the atmosphere by chemical reactions to form nitric acid, alkyl nitrates, and peroxynitrates, and by dry deposition of $NO_2$ (Crutzen, 1979; Jacob and Wofsy, 1990; Romer et

al. 2016). The chemical loss pathways of $NO_x$ have been extensively studied, but the physical loss of $NO_2$ to dry deposition remains much more uncertain. Globally, foliar deposition of $NO_2$ removes 20–50% of soil-emitted NO (Jacob and Wofsy,1990;Yienger and Levy, 1995), and constrains near-surface $NO_x$ concentrations and input to ecosystems (Hardacre et al. 2015). Understanding the processes that control this removal of $NO_x$ by the biosphere is important for predicting anthropogenic surface ozone and understanding flows in the nitrogen cycle.

Reactive nitrogen oxides also serve as an important nutrient in ecosystems. Exchange processes cycle nitrogen between the biosphere and atmosphere, influencing the availability of nitrogen to ecosystems (Townsend et al., 1996; Holland et al., 1997; Galloway et al., 2004; Holland et al., 2005). Deposition of atmospheric reactive nitrogen species can fertilize ecosystems with limited nitrogen availability (Ammann et al., 1995; Townsend et al., 1996; Williams et al., 1996; Holland et al., 1997; Galloway et al., 2004; Teklemariam and Sparks, 2006). Although nitrogen is often the limiting nutrient for plant growth (Oren et al., 2001; Galloway et al., 2004), anthropogenic activities have in some cases caused an excess loading of nitrogen to ecosystems, leading to dehydration, chlorosis, soil acidification, and a decline in productivity (Vitousek et al.,1997; Fenn et al., 1998; Galloway et al., 2004).

The current understanding of the exchange of nitrogen oxides between the atmosphere and biosphere remains incomplete. Despite the importance of dry deposition processes, they are among the most uncertain and poorly constrained aspects of atmosphere-biosphere nitrogen exchange and the tropospheric budgets of $O_3$ and $NO_x$ (Wild, 2007; Min et al., 2014; Hardacre et al., 2015). This uncertainty arises from the complex dependence of dry deposition processes on surface cover, meteorology, seasonal changes in leaf area index (LAI), species of vegetation, and the chemical species carrying odd-N. Developing a mechanistic understanding of dry deposition of $NO_2$ has largely depended on inferences from scarce long-term field observation data and a limited number of laboratory studies on the effects of environmental factors on deposition at the leaf-level. This understanding is represented by a deposition velocity, $V_d$. Many global scale chemical transport models (Wesely, 1989; Jacob and Wofsy, 1990; Ganzeveld and Lelieveld, 1995; Wang and Leuning, 1998; Ganzeveld et al., 2002a) parameterize $V_d$ using the resistance in-series approach similar to that developed by Baldocchi et al. (1987). These treatments are heavily parameterized, leading to a large degree of uncertainty, many of which (Jacob and Wofsy, 1990; Wesely, 1989) do not account for the effects of VPD, SWP, $CO_2$ mixing ratio, or other factors known to influence stomatal conductance (Hardacre et al., 2015). A common approach for modelling canopy uptake of trace gases is with a one- or two- layer "big-leaf" dry deposition model, in which the forest is treated as having a characteristic "average" deposition velocity (Hicks et al. 1987; Wesely, 1989; Ganzeveld and Lelieveld 1995; Wang and Leuning, 1998; Zhang et al., 2002). However, Ganzeveld et al. (2002b) implemented a multi-layer column model in a global chemistry and general circulation model GCM-ECHAM (European Centre Hamburg Model) to study the role of canopy interactions in global atmosphere-biosphere $NO_x$ exchange and demonstrated the importance of considering interactions within the canopy, particularly in pristine forest sites. More comprehensive treatments of atmosphere-biosphere exchange are thus needed in global models.

The deposition velocity of $NO_2$ to vegetation is largely regulated by stomatal conductance (Johansson, 1987; Thoene et al., 1991; Rondon and Granat, 1994; Teklemariam and Sparks, 2006; Chaparro-Suarez et al., 2011; Breuninger et al., 2012; Delaria et al., 2018), which varies with tree species, photosynthetically active radiation (PAR), vapor pressure deficit (VPD), temperature (T), soil water potential (SWP) and seasonality of leaf phenology (Emberson et al., 2000; Zhang et al., 2003; Altimir et al., 2004; Hardacre et al., 2015; Kavassalis and Murphy, 2017). $NO_2$ deposition remains even more uncertain than deposition of $O_3$, where stomatal response has been shown to be the primary regulator of foliar deposition and mesophyllic resistance to deposition is negligible. Observations from leaf-level laboratory studies suggest the deposition of $NO_2$ is also

controlled by stomatal aperture (Hanson and Lindberg, 1991; Rondon and Granat, 1994; Hereid and Monson, 2001; Teklemariam and Sparks, 2006; Pape et al., 2008; Chaparro-Suarez et al., 2011; Breuninger et al., 2012; Delaria et al., 2018), however, reactions in the mesophyll may also be important for controlling the deposition velocity of $NO_2$ (Teklemariam and Sparks, 2006; Breuninger et al., 2012). A failure to consider the effects of relevant meteorology on stomatal conductance, as well as our deficient understanding of mesophyllic resistances and the diversity of ecosystem responses, severely limits our ability to understand dry-deposition processes and how they will be affected by feedbacks from changes in climate, land use, and air pollution.

The importance of these considerations has recently been illustrated by Kavassalis and Murphy (2017), who found a significant correlation between VPD and ozone loss, and demonstrated that modeling using VPD-dependent parameterizations of deposition better predicted the correlation they observed. Previous work by Altimir el al. (2004) and Gunderson et al. (2002) have described the effects of VPD and other environmental parameters on the stomatal conductance to $O_3$ of *Pinus sylvestris* and *Liquidambar styraciflua*, respectively. More recent models, like the DO3SE model for estimating stomatal conductance to predict ozone deposition velocities, fluxes and damage to plants, incorporate the effects of VPD and SWP on stomatal conductance. No similar model exists for assessing these effects on $NO_x$ deposition, although Ganzeveld et al. (2002b) included the effect of soil moisture availability for evaluating the role of canopy $NO_x$ uptake on canopy $NO_x$ fluxes. The DO3SE has successfully been implemented in the European Monitoring and Evaluation Program (EMEP) regional model (2012). Modelling studies by Buker et al. (2007) and Emberson et al. (2000) have also demonstrated the success of regional-scale parameterizations using observed relationships between meteorology and stomatal conductance for application to $O_3$. Such treatments of VPD and SWP were incorporated into a regional air quality model by Zhang et al. (2002 and 2003).

In this study we present a simplified multi-layer atmosphere-biosphere exchange model and investigate the sensitivity of $NO_x$ canopy fluxes, ozone production, $NO_x$ vertical profiles, and $NO_x$ lifetimes to different parameterizations of stomatal conductance and deposition velocity. We consider here both the Wesely model and the similarly simplistic approach of Emberson et al. (2000) that incorporates effects of VPD and SWP. We restrict our considerations to the effects of different stomatal resistance parameterizations on predicted deposition velocities, as the magnitude of the mesophyllic resistance remains uncertain and is assumed to be comparatively small in atmospheric models (Zhang et al., 2002). We also restrict our considerations to $NO_2$ deposition, as NO deposition has been shown to be negligible in comparison (Delaria et al., 2018). There have been many studies investigating the effects of dry-deposition parameterizations on deposition velocities—particularly of ozone—and the abilities of different modeling schemes to reproduce observational data for other molecules such as $NO_2$, NO, $H_2O_2$, $HNO_3$, hydroxy nitrates, alkyl nitrates, peroxyacyl nitrates, etc. (Zhang et al., 1996; Wang et al., 1998b; Emberson et al., 2000; Ganzeveld 2002a; Buker et al., 2007; Wolfe et al., 2011; Hardacre et al., 2015; Nguyen et al., 2015). However, there has been little evaluation of how changes in dry deposition of $NO_2$ may affect surface mixing ratios and chemistry of important atmospheric species. Assessing the sensitivity to $NO_2$ deposition is crucial not only for evaluating the potential impact of uncertainties in dry-deposition parameterizations for global and regional models, but for understanding how a changing climate may influence $NO_x$, surface ozone, and the nitrogen cycle.

## 2 Model description

We have constructed a simple atmospheric model for investigating the influence of leaf-level $NO_2$ foliar deposition on canopy scale $NO_x$ lifetimes and concentrations. The model consists of three canopy layers and a total of eight vertical boxes within the planetary boundary layer (PBL), taken to be 1000 m during the day and 60 m at night (Wolfe and Thornton, 2011; Wolfe et al., 2011). The increase in PBL height during the day is treated as a Gaussian function of time with 98% of the integrated area contained between sunrise and sunset, with the maximum height reached at the time of maximum daily temperature (Fig.1). The first two boxes above the canopy were kept at a constant altitude, as the evolution of these layers was found to have a minimal effect on the model results discussed. The model was designed to be representative of a homogenous forest environment with the aim of simulating observations at forest tower sites.

In each box, the change in concentration ($C$) of species $i$, is calculated using the time-dependent continuity equation:

$$\frac{\partial C_i(z)}{\partial t} = P(z) + L(z) + E(z) + D(z) + A(z) + \frac{\partial F(z)}{\partial z} \tag{1}$$

where the terms on the right are the chemical production, chemical loss, emission, deposition, advection, and turbulent flux, respectively. In each box ($k=1-8$) the altitude (z) is considered as the average of the altitudes at the upper boundaries of boxes $k$ and $k-1$ (the midpoint of box $k$). The change in concentration for species $i$ is calculated for each time step $\Delta t = 2$ s (Table 1).

$$\Delta C_{i,k} = \left( P_{i,k} + L_{i,k} + E_{i,k} + D_{i,k} + A_{i,k} + \frac{F_{i,k}}{\Delta h_k} \right) \Delta t \tag{2}$$

where $\Delta h_k$ is the width of box $k$. The only species not treated in this manner is the hydroxyl radical (OH), which is calculated using a steady-state approximation.

### 2.1 Deposition

The deposition flux ($F_{dep}$) of each depositing species $i$ in the canopy is calculated according to:

$$F_{dep} = -V_d \cdot LAI \cdot C_i \tag{3}$$

where LAI is the leaf area index, and $V_d$ is the deposition velocity. The deposition velocities are calculated according to:

$$V_d = \frac{1}{R} \tag{4}$$

where R is the total resistance to deposition.

$$R_{leaf} = \left( \frac{1}{R_{cut}} + \frac{1}{R_{st}+R_m} \right)^{-1} \tag{5}$$

$$R = R_a + R_b + R_{leaf} \tag{6}$$

where $R_a$, $R_b$, $R_{cut}$, $R_{st}$, and $R_m$ are the aerodynamic, boundary layer, cuticular, stomatal, and mesophilic resistances, respectively. These resistances describe the turbulent transport of a gas to the surface ($R_a$), molecular transport through a thin layer of air above the leaf surface ($R_b$), and deposition to the leaf surface ($R_{leaf}$) (Baldocchi et al., 1987). $R_{leaf}$ is dependent upon plant physiology and the chemical and physical properties of the deposition compounds. $R_{leaf}$ is determined by

deposition to the leaf cuticles ($R_{cut}$), diffusion through the stomata ($R_{st}$), and chemical processing within the mesophyll ($R_m$). We do not allow for emission of NO or NO$_2$ from leaves, consistent with recent laboratory observations that have observed negligible compensation points for these molecules (Chaparro-Suarez et al., 2011; Breuninger et al., 2013; Delaria et al., 2018).

All boundary, aerodynamic, cuticular, and soil resistances of O$_3$, HNO$_3$, CH$_2$O, alkyl nitrates (ANs) and acylperoxy nitrates (APNs), HC(O)OH, ROOH, and H$_2$O$_2$ are calculated according to Wolfe et al. (2011). The cuticular and mesophyllic resistances for NO$_2$ and NO are adjustable input parameters. Stomatal resistances are determined from the stomatal conductance to water vapor ($g_s$) calculated using either Eq. 7 (Wesely, 1989), or Eq. 8 (Jarvis et al., 1976; Emberson et al., 2000), hereafter referred to as the Wesely and Emberson schemes, respectively:

$$g_s = g_{max} \times \frac{T(40-T)/400}{(1+(200(SR+0.1)^{-1})^2)} \tag{7}$$

$$g_s = g_{max} \times f_{phen} \times f_{light} \max\{f_{min}, (f_{temp} \times f_{VPD} \times f_{SWP})\} \tag{8}$$

where $g_{max}$ is the species-specific maximum stomatal conductance, $f_{min}$ is a species-specific scaling factor to the minimum stomatal conductance, SR is the solar radiation in W m$^{-2}$, and $f_{phen}$, $f_{SWP}$, $f_{light}$, $f_{temp}$, and $f_{VPD}$ are functions representing modifications to the stomatal conductance due to leaf phenology, soil water content, irradiance, temperature, and vapor pressure deficit, respectively (Eq. 9–12).

$$f_{light} = 1 - \exp(-Light_a \times PPFD) \tag{9}$$

$$f_{temp} = 1 - \frac{(T-T_{opt})^2}{(T_{opt}-T_{min})^2} \tag{10}$$

$$f_{VPD} = \min\left\{1, \left((1-f_{min}) \times \frac{(VPD_{min}-VPD)}{(VPD_{min}-VPD_{max})}\right) + f_{min}\right. \tag{11}$$

$$f_{SWP} = \min\left\{1, \left((1-f_{min}) \times \frac{(SWP_{min}-SWP)}{(SWP_{min}-SWP_{max})}\right) + f_{min}\right. \tag{12}$$

T$_{opt}$ and $T_{min}$ are the optimal and minimum temperature required for stomatal opening. PPFD is the photosynthetic photon flux density and $Light_a$ is a species-specific light response parameter. $VPD_{min}$ and $VPD_{max}$ are the vapor pressure deficit at which stomatal opening reaches a minimum and maximum, respectively. $SWP_{min}$ and $SWP_{max}$ are the soil water potentials at which stomatal opening reaches a minimum and maximum, respectively. All model calculations represented the peak growing season when $f_{phen}= 1$. $f_{temp}$, $f_{VPD}$, and $f_{light}$ were calculated according to Emberson et al. (2000) using parameters found in Table 2.

## 2.2 Site description

The model was evaluated with comparison to observations from the Biosphere Effects on Aerosols and Photochemistry 2009 (BEARPEX-2009) field campaign from 15 June – 31 July 2009 at Blodgett forest (Min et al., 2014), and the University of Michigan Biological Station (UMBS) during 5 August – 10 August 2012 (Geddes and Murphy, 2014). For

the BEARPEX-2009 calculations, the modelled canopy included an overstory height of 10 m with a one-sided leaf area index (LAI) of 3.2 $m^2m^{-2}$ ($LAI_{os}$), and an understory height of 2 m with a LAI of 1.9 $m^2m^{-2}$ ($LAI_{us}$). Model simulations were run for June 30, 2009 using conditions from the BEARPEX-2009 ponderosa pine forest site located in the western foothills of the Sierra Nevada Mountains, CA (38°58'42.9"N, 120°57'57.9"W, elevation 1315 m) (Table 1) (Fig. 2a). Meteorological conditions and soil NO emissions used in the model simulation were those reported by Min et al. (2014). Diurnal soil water potentials (SWP) were values reported in a geological survey of nearby Sierra sites in a comparatively wet year (Ishikawa and Bledsoe, 2000; Stern et al., 2018).

For UMBS-2012 calculations, the modelled canopy included an overstory height of 20 m with a one-sided LAI of 2.5 $m^2m^{-2}$, and an understory height of 4 m with a LAI of 1 $m^2m^{-2}$ (Bryan et al. 2015). Model simulations were run for August 8, 2012 using conditions from the UMBS mixed hardwood forest located in northern Michigan (45°33'32" N, 84°42'52"W) (Table 1) (Fig 2b). Daily temperatures, VPDs, soil NO emissions and site-specific parameters used in the model simulations were those reported in Geddes and Murphy (2014), and Seok et al. (2013).

Temperature and relative humidity used in the model were sinusoidal fits to observations of minimum and maximum daily temperature and relative humidity from the corresponding field measurement site. The relative temperature decrease as a function of altitude was calculated using a fit to observations during BEARPEX-2007, as presented by Wolfe and Thornton (2011). Solar zenith angles (SZA) and photosynthetically active radiation (PAR) were calculated every 0.5 h for each location and time period using the National Center for Atmospheric Research TUV calculator (Madronich and Flocke, 1999) and fit using a smoothed spline interpolation. Within the canopy, extinction of radiation ($ER$) was calculated following Beer's law:

$$ER_k = \exp\left(-\frac{k_{rad}LAI_{cum}}{cos(SZA)}\right) \tag{13}$$

where $k_{rad}$ is the radiation extinction coefficient, $SZA$ is the solar zenith angle, and $LAI_{cum}$ is the cumulative LAI calculated as the sum of one-half the LAI in box $k$ and the total LAI in the boxes above box $k$.

## 2.3 Vertical transport and advection

The turbulent diffusion flux ($F(z)$) is represented in the model using K-theory, according to the Chemistry of Atmosphere-Forest Exchange (CAFE) Model (Wolfe and Thornton, 2011).

$$F(z) = -K(z)\frac{\Delta C_{i,k}}{\Delta z} \tag{14}$$

where $\Delta C_{i,k}$ is the change of concentration in species $i$ in box $k$ during each timestep and $\Delta z$ is the difference between the midpoints of boxes $k$ and $k+1$. $K(z)$ above the canopy is based on the values from Gao et al. (1993) and below is a function of friction velocity calculated according to Wolfe et al. (2011) and is a function of the diffusion timescale ratio ($\tau/T_L$)— defined as the ratio of the "time since emission" of a theoretical diffusing plume ($\tau$) and the Lagrangian timescale ($T_L$)—and the friction velocity ($u^*$) (Wolfe and Thornton, 2011). The details of the parameterization of turbulent diffusion fluxes is documented elsewhere (Wolfe and Thornton, 2011) and based on the works of Raupach (1989) and Makar et al. (1999). The

height dependent friction velocity $(u(z)^*)$ is attenuated from the above-canopy $u^*$ according to Yi et al. (2008). Although Finnigan et al. (2015) identified flaws in this treatment, we believe it is sufficient for our focus on illustrating generalizable qualitative trends.

The resulting residence time in the canopy is approximately 2–3 min for model conditions during the day. Our model is a simple parameterization of turbulent processes and as such will only capture mean vertical diffusion. Other works (Collineau and Brunet, 1993a; Raupach et al., 1996; Brunet and Irvine, 2000; Thomas and Foken, 2007; Sörgel et al., 2011; Steiner et al., 2011) have shown that "near-field" effects of individual canopy elements and coherent turbulent structures can play an important role in canopy exchange. These more intricate processes are not captured explicitly by our simple model. Previous work (Gao et al., 1993; Makar et al., 1999; Stroud et al., 2005; Wolfe et al., 2011) have also utilized fairly simple representations of canopy exchange in local and regional models, and as such K-theory is likely sufficient to represent average vertical diffusion for the purposes of our study.

Advection in the model is treated as a simple mixing process in each model layer.

$$\left(\frac{dC_i}{dt}\right) = -k_{mix}\left(C_i - C_{i(adv)}\right) \tag{15}$$

where $k_{mix} = 0.3$ h$^{-1}$ (Wolfe and Thornton, 2011), and $C_{i(adv)}$ is the advection concentration of species $i$. Advection concentrations are set to fit with the observations during BEARPEX-2009 (Min et al., 2014) or UMBS-2012 (Geddes and Murphy, 2014; Seok et al., 2013) and are used to maintain reasonable background concentrations (Table S1). Concentrations of $NO_x$, $O_3$, and some VOCs at both sites were influenced by emissions from nearby cities and consequently had sources outside the canopy. For the BEARPEX-2009 model runs, the maximum daily advection concentration was reached at around 17 hrs, based on field observations of higher $NO_x$ plumes from near-by Sacramento in the afternoon (Wolfe et al., 2011; Min et al., 2014). The diurnal advection concentrations of $NO_x$ were modelled with a sinusoidal function in the range 0.1-0.35 ppb (Table S1). For UMBS all advection concentrations were constant.

## 2.4 Chemistry

Chemistry in the model is based on reaction rate constants from the JPL Chemical Kinetics and Photochemical Data Evaluation No. 18 (Burkholder et al., 2015). Photolysis rates are calculated as a function of solar zenith angle (SZA), which was constructed using a smoothed spline interpolation fit of photolysis rates calculated with the TUV calculator (Madronich and Flocke, 1999) at every ten-degree interval of the zenith angle. The simplified reaction scheme included in the model is based on the model presented in Browne and Cohen (2012). The model includes both daytime and night-time $NO_x$ chemistry and a simplified oxidation scheme. In this simplified case, oxidation of volatile organic compounds (VOCs) during the daytime results in the production of peroxy radicals ($RO_2$), treated as a uniform chemical family. To be applicable to a range of forest types, we also include adjustable parameters, *kOH* and *kNO₃* for the average rate constant for reaction of VOC with OH and

NO$_3$, respectively. kOH and kNO$_3$ are effective values adjusted in the model based on site-specific VOC composition and observations of OH reactivity. A complete list of reactions and rate constants included in the model is shown in Table S2.

## 2.5 BVOC emissions

Emissions rates (molecules cm$^{-3}$s$^{-1}$) of biogenic volatile organic compounds (BVOCs) in the canopy are calculated via:

$$E(z) = \frac{E_b}{\Delta h} C_L(z) C_T(z) LAI \tag{16}$$

where $E_b$ (molecules cm(leaf)$^{-2}$ s$^{-1}$) is the basal emission rate of VOC, $\Delta h$ is the total height of the box, and $C_L$ and $C_T$ are corrections for light and temperature (Guenther et al., 1995).

## 2.6 Evaluation of NO$_x$ fluxes and lifetimes

The model was used to assess the impact of NO$_2$ deposition parameters on the NO$_x$ budget, lifetime, loss, and vertical profile within a forested environment. In each box, the rates of NO$_x$ loss with respect to nitric acid formation, alkyl nitrate formation, and deposition were calculated from Eq. 17–19.

$$L_{NO_x \rightarrow HNO_3} = k_{OH+NO_2}[OH][NO_2] + k_{N_2O_5\ hydrolosis}[N_2O_5] + k_{NO_3+aldehyde}[aldehyde][NO_3] \tag{17}$$

$$L_{NO_x \rightarrow RONO_2} = \alpha k_{NO+RO_2}[NO][RO_2] + \beta k_{NO_3}[NO_3][BVOC] \tag{18}$$

$$L_{NO_x \rightarrow Dep} = F_{dep}/\Delta h_k , \tag{19}$$

$\alpha$ is the fraction of the NO + RO$_2$ reaction that forms alkyl nitrates and $\beta$ is the fraction of the NO$_3$ + BVOC reaction that forms alkyl nitrates. The NO$_x$ lifetime was then scaled to the entire boundary layer by summing over the products of the lifetime and boundary layer fraction ($\Delta h_k/PBL$) in each box

$$\tau_{PBL} = \frac{\sum_{k=1}^{8}[NO_x]_k}{\sum_{k=1}^{8}(L_{NO_x \rightarrow Dep} + L_{NO_x \rightarrow RONO_2} + L_{NO_x \rightarrow HNO_3})} \tag{20}$$

NO$_x$ was treated as the sum of NO, NO$_2$, and all short-lived products, including NO$_3$, 2N$_2$O$_5$, and peroxyacetyl nitrate (PAN) (Romer et al., 2016). Deposition of PAN was not considered.

We also calculated the 24 h average vertical fluxes (Eq. 14) of NO$_x$, and used the flux through the canopy to estimate the fraction of soil emitted NO$_x$ ventilated to the troposphere above. Because PAN formed during the nighttime is expected to re-release NO$_x$ to the atmosphere during the day, in this calculation, PAN was included as part of the NO$_x$ budget.

## 3 Sensitivity to parameterizations

We assessed the sensitivity of the model to $\tau/T_L$, the radiation extinction coefficient ($k_{rad}$), the aerodynamic leaf width ($l_w$), LAI, soil NO emission (*eNO*), and $\alpha$. These parameters are simplifications of complex physical processes and not always easily constrained by observations. The total deposition velocity of NO$_2$ chosen for these assessments was 0.2 cm s$^{-1}$ during the daytime and 0.02 cm s$^{-1}$ during the nighttime, based on values of g$_{max}$ and g$_{min}$ chosen for Blodgett forest (discussed above)

and typical values for deposition velocity observed for a variety of species in the laboratory (Teklemariam and Sparks, 2006; Chaparro Suarez et al., 2011, Breuninger et al., 2013, Delaria et al., 2018).

The largest effects were observed for changes in $\alpha$, LAI, and soil NO emission. $LAI_{os}$ and $LAI_{us}$ were scaled from their values of 1.9 m$^2$/m$^2$ and 3.2 m$^2$/m$^2$, respectively by a factor of 0.25 and 1.5. Increasing the scaling factor from 0.25 to 1.5 resulted in a decrease of NO$_x$ lifetimes, above canopy concentration, and average canopy flux of 24%, 27%, and 36%, respectively (Fig. S1). Increasing $\alpha$ from 0.01 to 0.1 resulted in a decrease in NO$_x$ lifetimes, above canopy concentrations, and average canopy fluxes of 75%, 38%, and 39%, respectively (Fig. S2). For all other model runs an $\alpha$ of 0.075 was chosen, in accordance with observations from regions primarily influenced by BVOCs (eg. monoterpenes, isoprene, 2-methyl-3-buten-2-ol). Increasing the maximum soil NO emission from 1 to 10 ppt m s$^{-1}$ increased the in-canopy enhancement from 28% to 140% relative to above-canopy NO$_x$ concentrations (Fig. S3b). The fraction of soil-emitted NO$_x$ ventilated through the canopy also increased from 45% to 64% (Fig. S3a). The large effect of soil NO emission on NO$_x$ fluxes implies that this highly variable parameter (Vinken et al., 2014) is also important to constrain in chemical transport models. Further discussion of soil NO emission is, however, beyond the scope of this study.

Very small effects on NO$_x$ were observed for changes in the parameters $\tau/T_L$, $k_{rad}$, or $l_w$. The minor changes caused by variations in these parameters are listed below for completeness:

$\tau/T_L$ represents the diffusion timescale ratio, a full description of which can be found in Wolfe and Thornton (2011). Larger $\tau/T_L$ represents faster diffusion and vertical transport within the canopy layer, and shorter residence times in the canopy. We find that altering this parameter from 1.2 to 8 (representing a change in residence time from 650 s to 62 s) caused a 9.9%, 4.4%, and 8.7% increase in average canopy fluxes, NO$_x$ lifetimes and above canopy concentration, respectively (Fig. S4). For all subsequent model runs, a value of 2 for $\tau/T_L$ was chosen, resulting in a canopy residence time during the day of 152 s and 194s for Blodgett Forest and UMBS, respectively, calculated using Eq.21.

$$\tau_{can} = h_{can} \sum_{k=1}^{3} \frac{\Delta h_k}{K(z_k)} \tag{21}$$

The boundary layer resistance, or laminar sublayer resistance, $R_b$, is dependent upon the aerodynamic leaf width, $l_w$ (Eq.22)

$$R_b = \frac{cv}{Du^*(z)} \left( \frac{l_w u^*(z)}{v} \right)^{1/2} \tag{22}$$

where $v = 0.146$ cm$^2$ s$^{-1}$ is the kinematic viscosity of air, $D$ is the species-dependent molecular diffusion coefficient, $c$ is a tunable constant set to 1 for this study, and $u^*(z)$ is the height-dependent friction velocity that is a function of $u^*$ and $LAI_{cum}$ (Wolfe and Thornton, 2011). $l_w$ depends upon the vegetation species. A value of 1 cm was chosen for the overstory and 2 cm for the understory, as these widths are characteristic of pine trees and understory shrubs in a poderosa pine forest (Wolfe and Thornton, 2011). Species with rapid deposition to the cuticles or the stomata are expected to be more sensitive to errors in $l_w$, such as HNO$_3$ or H$_2$O$_2$. An increase in NO$_x$ lifetime, average canopy flux, and above canopy concentration of 1.4%, 2.4%, and 2.8%, respectively, was predicted for a change in $l_w$ scaling factor from 0.1 to 2 (Fig. S5). These changes are expected to be greater in forests with a larger average deposition velocity, where $R_b$ makes a greater contribution to the total resistance.

The rates of stomatal gas exchange and photolysis are regulated by the intensity of light that penetrates the canopy. The extinction of radiation by the canopy, treated as a Beer's Law parameterization (Eq. 113) is exponentially proportional to the radiation extinction coefficient, $k_{rad}$. $k_{rad}$ ranging from 0.4–0.65 has been measured for coniferous forests and understory shrubs (Wolfe and Thornton, 2011). The $NO_x$ lifetime increased by 2.7% and the canopy fluxes, and above-canopy concentrations decrease by 0.7% and 0.6%, respectively, for a change in $k_{rad}$ from 0 to 0.6 (Fig. S6). This effect is expected to be greater for forests with larger LAI. The minimal effect of $k_{rad}$ on model results was also observed for multiple canopy profile shapes of equivalent LAI.

## 4 Results

### 4.1 Model validation: comparison to field observations

To evaluate the applicability of our 1D multilayer canopy model for predicting $NO_x$ concentrations and vertical fluxes in a variety of forest environments, we compared the model to observations from BEARPEX-2009 and UMBS-2012. Parameters used in each calculation are shown in Table 1. The model was run using both the Emberson and Wesely stomatal conductance models. Parameters for temperature, drought stress, and maximum and minimum stomatal conductances used in the Emberson model were input for the dominant tree species in the region (Table 2). At the BEARPEX-2009 site, the dominant tree species was ponderosa pine. For this site, $g_{max}$ and parameters for $f_{SWP}$ and $f_{VPD}$ were inferred from ponderosa pine stomatal conductance data (Kelliher et al., 1995; Ryan et al., 2000; Hubbard et al., 2001; Johnson et al., 2009; Anderegg et al., 2017), and $f_{light}$ was inferred from measurements of the canopy conductance during BEARPEX-2009 (Fig 3a). $f_{temp}$ was represented by observations for Scots pine (Altimir et al., 2004; Emberson et al., 1997; Buker et al., 2012) and validated with comparison to stomatal conductance measured via sap-flow during BEARPEX-2009 (Fig 3a). At UMBS the dominant species are quaking aspen and bigtooth aspen, with many birch, beech, and maple species also present (Seok et al., 2013). Data for a European beech tree species was used to represent stomatal conductance parameters (Buker et al., 2007; Buker et al., 2012) and SWP stress (Emberson et al., 2000). These parameters were validated with comparison to stomatal conductance calculated from water vapor and latent heat flux measurements during UMBS-2012 using an energy-balance method according to Mallick et al. (2013) (Fig 4a).

The model replicates key features of the canopy fluxes and above-canopy $NO_x$ daytime mixing ratios from the 2009 BEARPEX campaign (Fig.3). The average daytime above-canopy $NO_x$ mixing ratios during the duration of BEARPEX-2009 was 253 ppt, with observations ranging from 80–550 ppt of $NO_2$ and 10–100 ppt of NO (Min et al., 2014). The general daily trends in observations of $NO_x$ mixing ratios are captured by both the Wesely and Emberson cases—with minimum $NO_x$ mixing ratios occurring in the late morning, an increase of $NO_x$ in the afternoon, and maximum $NO_x$ concentrations of 450–500 ppt reached in the evenings, primarily as a result of high-$NO_x$ plumes from near-by Sacramento in the afternoon (Wolfe et al., 2011; Min et al., 2014) (Fig. 3b). However, both model scenarios predict a slower than observed decrease in $NO_x$ mixing ratios

from the evening to the early morning, larger mid-morning fluxes than observed (by ~0.5–1.5 ppt m s$^{-1}$), and fail to represent the in-canopy enhancement of NO$_x$ (~50 ppt), relative to above-canopy mixing ratios, observed in the evening (Fig 3). The above-canopy vertical NO$_x$ flux predicted in both model cases also agrees reasonably well with observations, with the Emberson case representing morning and midday NO$_x$ fluxes slightly better than the Wesely case. This relatively good agreement between the Emberson case and observed fluxes is also demonstrated in Fig 3d by the agreement between modelled and observed canopy NO$_x$ enhancements. There is, however, generally little difference between Emberson and Wesely model cases for this site during the period considered (Fig 3). This is likely due to the good agreement in both the Emberson and Wesely cases to observations of stomatal conductance (Fig 3a).

We also observe similar correspondence between the model and key features of the UMBS-2012 observations (Fig 4). NO and NO$_2$ mixing ratios and canopy fluxes are both within the range of observations. The model predicts a maximum of ~40% lower NO$_2$ in the morning and ~30% higher NO$_2$ at night than what was observed (Fig 4b). It should also be noted that this agreement was achieved without inclusion of an NO$_2$ compensation point, whereas Seok et al. (2013) had proposed the importance of considering foliar NO$_2$ emission at this location. Differences between the Wesely model and Emberson model were negligible for this site. This is likely due to a higher humidity in the summer in this region and larger soil moisture, reducing the prediction for midday and late afternoon VPD stress by the Emberson model, as can be seen by the similarity in the predicted g$_s$ by the Emberson and Wesely models (Fig 4a).

## 4.2 Effects of maximum stomatal conductance

The BEARPEX-2009 case was simulated using the Wesely model for different values of the maximum stomatal conductance ($g_{max}$) (Fig 5), with advection concentrations of NO$_x$ set to zero. The range of $g_{max}$ currently represented in the literature during peak growing season for forested regions ranges from 0.2–0.8 cm s$^{-1}$ (Kelliher et al., 1995; Emberson et al., 1997; Emberson et al., 2000; Ryan et al., 2000; Hubbard et al., 2001; Altimir et al., 2003; Fares et al., 2013). This range reflects differences in forest types and a wide variety of tree species. Global CTMs using the Wesely parameterization currently include $g_{max}$ of 1.4, 0.77, and 1 cm s$^{-1}$ for deciduous, coniferous, and mixed forests, respectively (Wesely, 1989; Wang et al., 1998a). Figure 5b demonstrates the impact of $g_{max}$ on the average daily vertical flux of NO$_x$ through the canopy. 96% of soil emitted NO$_x$ is ventilated through the canopy with no foliar deposition ($g_{max}$= 0 cm s$^{-1}$). In contrast, 44% of soil-emitted NO$_x$ is taken up by the forest and 56% ventilated through the canopy when the maximum stomatal conductance is 1.4 cm s$^{-1}$. Figures 5c and 5d show the effects of $g_{max}$ on the diurnal flux through the canopy and the diurnal above canopy NO$_x$ mixing ratio, respectively. Compared with no foliar deposition, a $g_{max}$ of 1.4 cm s$^{-1}$ results in ~60% reduction in the canopy flux and ~50% reduction in the above-canopy NO$_x$ mixing ratio at noon. (Fig. 5c, d). In Figure 6a we show the fraction of soil-emitted NO$_x$ ventilated through the canopy as a function of $g_{max}$. The model suggests a maximum foliar reduction of NO$_x$ of ~60% for a canopy of 10 m and total LAI of 5.1 m$^2$/m$^2$. Our model also predicts that changes in $g_{max}$ have a greater overall impact on canopy NO$_x$ fluxes at larger leaf resistances and slower foliar uptake. In the range for $g_{max}$ of ~0–0.5 cm s$^{-1}$, variation in $g_{max}$ can have a large impact on the predicted canopy fluxes of NO$_x$, which would in turn have a large impact on concentrations and

fluxes of $O_3$. These values of $g_{max}$ results in deposition velocities in the range expected for most forests, based on laboratory measurements of leaf-level deposition (Hanson and Lindberg, 1991; Rondon and Granat, 1994; Hereid and Monson, 2001; Teklemariam and Sparks, 2006; Pape et al., 2008; Chaparro-Suarez et al., 2011; Breuninger et al., 2013; Delaria et al., 2018) and global analysis suggesting 20–50% reductions in soil-emitted $NO_x$ by vegetation (Jacob and Wofsy, 1990; Yienger and

Levy, 1995, Ganzeveld et al., 2002a and 2002b). Model calculations also predict a strong effect on the lifetimes of $NO_x$, as shown in Figure 6b, with maximum stomatal conductances of 0.1 cm s$^{-1}$, 0.3 cm s$^{-1}$, and 1.4 cm s$^{-1}$ reducing the $NO_x$ lifetime by ~ 0.7 hrs (~7%), ~1.8 hrs (~18%), and ~3.6 hrs (~36%), respectively compared with no deposition. Similar trends (not shown) were also observed using parameters for UMBS.

## 4.3 Emberson model vs. Wesely model comparison

The relative importance of including parameterizations of VPD and SWP in the calculation of stomatal conductance and overall deposition velocity is expected to be regionally variable, along with regional variations in dominant tree species, soil types, and meteorology. We ran the model using BEARPEX-2009 conditions using both the Wesely and Emberson stomatal conductance models under "dry" and "wet" conditions. Here we use "dry" to refer to conditions of low humidity and low soil moisture and "wet" to refer to conditions with high humidity and high soil moisture. Under the "dry" scenario the SWP daily

minimum and maximum were -2.0 MPa and -1.7 MPa, respectively, with the daily minimum reached at sunset. A minimum daily RH of 40% occurred at noon, with a maximum at midnight of 65%. Summertime is often even drier in regions of the western United States, so these "dry" parameters are conservative estimates for many forests. Under the "wet" scenario the SWP daily minimum and maximum were -0.5 MPa and -0.1 MPa, respectively. The maximum and minimum RH were 90% and 80%, respectively. The values for soil moisture and relative humidity chosen were based on observations of SWP by

Ishikawa and Bledsoe (2000) and the long-term climate data record at Auburn Municipal Airport (38.9547° N, -121.0819° W) from NOAA National Centers for Environmental Information.

The results of the Wesely and Emberson "wet" and "dry" model runs are shown in Figure 8. There was only a slight decrease of the in-canopy $NO_x$ enhancement and the canopy flux in the Wesely "wet" case, presumably due to a slight increase in OH radicals at higher RH. Predictably, the difference in the modelled deposition velocities was quite dramatic between the

Emberson "wet" and "dry" cases. In the "dry" scenario, the deposition velocity reached a maximum in the late morning, but rapidly declined after noon. The maximum deposition velocity reached was also substantially reduced (Fig 7a). Using the "wet" Emberson stomatal conductance model, the $NO_x$ flux out of the forest was reduced by 16% midday compared to the "dry" case, and the percent of soil $NO_x$ removed within the canopy was increased from 18% to 30% (Fig 7). The model calculates a substantial impact on above-canopy $NO_x$ mixing ratios (Fig. 8), with a maximum of ~30% difference in $NO_x$ in

the afternoon between "wet" and "dry" days using the Emberson parameterization, compared with ~10% difference using the Wesely model. Using the Emberson parameterization of stomatal conductance, deposition during "wet" days is predicted to contribute substantially more to the total $NO_x$ loss (~40%), with only ~15% contribution predicted for "dry" days (Fig. 9).

Under the Wesely model, where stomatal conductance is parameterized only with temperature and solar radiation, the predicted deposition velocity would be nearly identical between the spring and fall in the western United States and similar semi-arid regions (with comparatively minor temperature effects). While the Emberson model predicts large seasonal differences, the Wesely model fails to account for the dramatic decrease in stomatal conductance seen in the dry seasons in such regions caused by significant reductions in relative humidity and soil water potential (Prior et al., 1997; Panek and Goldstein, 2001; Chaves, 2002; Beedlow et al., 2013). We recognize that the multibox model presented in this work is a simplified representation of physical processes, and as such is not likely to (and is not intended to) provide quantitative exactitude for the trends described above. However, we argue for the necessity of incorporating these conceptual advances for accurately representing canopy processes and predicting their effect on the $NO_x$ cycle.

## 5 Discussion

### 5.1 Implications for modelling $NO_2$ dry deposition

As in our multilayer canopy model, the most common current method of parameterizing stomatal and cuticular deposition in large-scale chemical transport models (CTMs) is through the resistance model framework of Baldocchi (1987). Many global (e.g. WRF-Chem and GEOS-Chem) and regional (e. g. MOZART and CAMx) CTMs calculate the stomatal component of the total deposition resistance using the representation of Wesely (1989), where stomatal conductance is dependent only on the type of vegetation, temperature, and solar radiation. The limitations of this parameterization have been highlighted by observations of a strong dependence of foliar deposition on soil moisture and vapor pressure deficit (VPD) (Kavassalis and Murphy, 2017; Rydsaa et al., 2016). Inadequate descriptions of vegetative species, soil moisture, drought stress, etc., can have a dramatic impact on model results, and result in significant discrepancies between models and observations (Wesely and Hicks, 2000). Failure to account for effects of plant physiology on deposition may result in misrepresentation of deposition velocities, which, as we demonstrate, can have a substantial impact on $NO_x$ lifetimes and mixing ratios above and within a forest canopy. This effect will be especially pronounced in areas, such as much of the western United States, where there are frequent periods of prolonged drought. Parameterizations of stomatal conductance, such as those presented in Emberson et al. (2000) and incorporated into some regional-scale CTMs (e.g. EMEP, MSC-W, and CHIMERE), if incorporated into global atmospheric models, could more accurately reflect the dependence of foliar deposition on meteorology and soil conditions. However, additional laboratory and field measurements on diverse plant species are also needed to determine appropriate, ecosystem-specific inputs to these parameterizations.

It should be noted that there have been significant recent advances in optimization approaches of stomatal modelling based on the theory that stomata maximize $CO_2$ assimilation per molecule of water vapor lost via transpiration (Medlyn et al., 2011; Bonan et al., 2014; Franks et al., 2017; Miner et al., 2017; Franks et al., 2018). Medlyn et al. (2011) reconciled the empirical widely utilized Ball-Berry model with a theoretical framework optimizing ribulose 1,5 bisphosphate (RuBP) regeneration-limited photosynthesis. However, such methods of water use efficiency optimization do not account for stomatal

closure as a result of soil moisture stress. Bonan et al. (2014) further developed a model considering water use efficiency optimization and water transport between the soil, plant, and atmosphere. Such parameterizations are utilized in the Community Land Model (CLM)—a land surface model often incorporated into regional and global climate-chemistry models (Lombardozzi et al., 2015; Kennedy et al., 2019). Although this model provides a physiological and mechanistic basis for

stomatal behaviour, it is heavily parameterized, relying on inputs of plant and soil parameters that could be expected to vary significantly across ecosystem types. For this reason, we view these methods as aspirational for incorporation into atmospheric global CTMs. We find the relative simplicity of the Emberson approach more useful for the purpose and scope of parameters for large-scale atmospheric models.

## 5.2 Implications for modelling ozone

$NO_2$, as well as $O_3$, deposition budgets are frequently calculated through inferential methods whereby the deposition velocity is constrained with ambient observations (Holland et al., 2005; Geddes and Murphy, 2014). These inferential models are often complicated by the fast reaction of the $NO_2$-$NO$-$O_3$ triad, making it difficult to separate chemical and physical processes. Further, these inferential models for determining dry deposition constrained with observations of chemical concentrations and eddy covariance measurements of fluxes are difficult to interpret because of similar chemical and turbulent

timescales (Min et al., 2014; Geddes and Murphy, 2014). Emission of $NO$ from soils, rapid chemical conversion to $NO_2$, and subsequent in-air reactions of $NO_x$ must be evaluated accurately in in order to correctly infer $NO_x$ and $O_3$ atmosphere-biosphere exchange from observations. Our multilayer canopy model applies a simple method of representing these processes and evaluating the separate effects of chemistry and dry deposition on the $NO_x$ budget in forests.

    Since the foliar deposition of $NO_2$ reduces the $NO_x$ lifetime and $NO_x$ that is transported out of the canopy, it will also

reduce the amount of ozone that is produced both within and above the canopy. Ozone production efficiency (OPE) in the canopy is calculated using Eq.23–25:

$$L(\mathrm{NO_x}) = L_{NO_x \to Dep} + L_{NO_x \to RONO_2} + L_{NO_x \to HNO_3}, \tag{23}$$

$$P(\mathrm{O_3}) = k_{HO_2+NO}[\mathrm{HO_2}][\mathrm{NO}] + k_{CH_3O_2+NO}[\mathrm{CH_3O_2}][\mathrm{NO}] + (1-\alpha)k_{RO_2+NO}[\mathrm{RO_2}][\mathrm{NO}], \tag{24}$$

$$\mathrm{OPE} = \frac{P(\mathrm{O_3})}{L(\mathrm{NO_x})}, \tag{25}$$

where $P(\mathrm{O_3})$ is the ozone production rate and $L(\mathrm{NO_x})$ is the $NO_x$ loss rate. The effect of stomatal conductance to $NO_2$ on OPE is shown in Figure 6c. An increase in $g_{max}$ from 0 to 0.3 cm s$^{-1}$ results in a decrease in OPE for the PBL from 24.0 to 20.7 (~14%), and a decrease to 17.0 (~30%) if $g_{max}$ is 1.4 cm s$^{-1}$. This is similar to OPE calculations that have been reported for forests and environments with $NO_x$ mixing ratios less than 1 ppb and heavily influenced by BVOC emissions (Marion et al., 2001;Browne and Cohen, 2012;Ninneman et al., 2017).

$NO_2$ deposition and the in-canopy chemistry of $NO_2$-$NO$-$O_3$ also impacts $O_3$ production and removal. $O_3$ deposition is frequently inferred from measurements of $O_3$ concentrations or eddy-covariance measurements (Wesely and Hicks, 2000; Kavassalis and Murphy, 2017). However, because $NO_2$ has a direct impact on ozone production, deposition of $NO_2$ can affect

inferences of $O_3$ deposition from observations. The 14% reduction of OPE and more than a 20% reduction in daytime $NO_x$ resulting from an increase in $g_{max}$ from 0 to 0.3 cm s$^{-1}$ can cause a parallel decrease in $O_3$ concentrations and fluxes independent of $O_3$ chemical loss or deposition. Thus, deposition of $NO_2$ must be taken into account when evaluating $O_3$ deposition losses from observed canopy fluxes.

## 5.3 Implications for near-urban forests

The analysis above suggests that the relative importance of chemical sinks and deposition will vary with $NO_x$ concentration. To evaluate the relative importance of $NO_2$ foliar deposition and chemistry as a function of $NO_x$ mixing ratio, a simplified 1-box model was also constructed with a simplified reaction scheme (Table S3), VOC reactivity of 8 s$^{-1}$, $\alpha$ of 0.075, and a $HO_x$ ($HO_x \equiv OH + HO_2$) production rate ($P_{HO_x}$) of $2\times10^6$ molecules cm$^{-3}$s$^{-1}$ (similar to conditions observed at BEARPEX-09). $RO_2$, OH, and $HO_2$ were solved for steady-state concentrations and $NO_x$ loss pathways were calculated via Eq. 26–29.

$$D_{NO_x} = LAI \cdot V_d \cdot \frac{h_{can}}{h_{PBL}}[NO_2] \tag{26}$$

where $h_{can}$ is the canopy height (15m), $h_{PBL}$ is the planetary boundary layer height (1000 m), and LAI is 5 m$^2$/m$^2$.

$$P_{HNO_3} = k_{OH+NO_2}[OH][NO_2] \tag{27}$$

$$P_{ANs} = \alpha k_{RO_2+NO}[RO_2]fNO \tag{28}$$

where

$$fNO = \frac{k_{RO_2+NO}[NO]}{k_{RO_2+NO}[NO]+k_{RO_2+HO_2}[HO_2]+k_{RO_2+RO_2}[RO_2]} \tag{29}$$

The results from this simplified box model are shown in Figure 9 and agree well with our 1D multi-box model near 10 ppb $NO_x$ (Fig S7). With deposition set to zero, nitric acid formation becomes a more significant sink of $NO_x$ than alkyl nitrate formation at around 1 ppb, and nitric acid formation accounts for greater than 70% of the total loss at 100 ppb. With a deposition pathway included, deposition acts as the dominant $NO_x$ sink above 30 ppb and at 10 ppb deposition and AN formation are each 20% of the $NO_x$ sink. Deposition is approximately 10% of the sink over a wide range of concentrations. Forests in close proximity to urban centers (Fig S9) may result in a substantial local decrease in $NO_x$ (Fig S8, Fig 10). Although the influence of urban or near-urban trees on $NO_x$ concentrations would be heavily dependent on meteorological factors (i.e. wind speed and direction), proximity to emission sources, and LAI, it may have some importance on a local or neighborhood scale. This effect may be relevant for understanding and predicting the effects of $NO_x$ reduction policies within and near cities. It may also be useful in considering as a direct nitrogen input to the biosphere, not mediated by soil processes.

## 6 Conclusions

We have constructed a 1D multi-box model with representations of chemistry and vertical transport to evaluate the impact of leaf-level processes on canopy-scale concentrations, lifetimes, and canopy fluxes of $NO_x$. Our model is able to closely replicate canopy fluxes and above-canopy $NO_x$ daytime mixing ratios during two field campaigns that took place in a Sierra Nevada

pine forest (BEARPEX-2009) and a northern Michigan mixed hardwood forest (UMBS-2012). We conclude that the widely used canopy reduction factor approach to describing soil $NO_x$ removal from the atmosphere within plant canopies is consistent with a process-based model that utilizes stomatal uptake and we recommend that the CRF parameter be replaced with stomatal models for $NO_2$ uptake.

5        We demonstrate with our 1D multi-box model that $NO_2$ deposition provides a mechanistic explanation behind canopy reduction factors (CRFs) that are widely used in CTMs. We predict a maximum of ~60% reduction in the fraction of soil-emitted $NO_x$ ventilated through the canopy when stomatal conductances are greater than 0.075 cm s$^{-1}$, consistent with the range of global CRFs used in current CTMs (Jacob and Wofsy, 1990;Yienger and Levy, 1995). Our model also predicts that changes in $g_{max}$ have a greater overall impact on canopy $NO_x$ fluxes at larger leaf resistances to uptake (slower foliar uptake). In the

range for $g_{max}$ of ~0–0.5 cm s$^{-1}$, errors or variability in stomatal conductance can have a large impact on the predicted canopy concentrations and fluxes of $NO_x$, which would in turn have large impact on concentrations and fluxes of $O_3$. This range of deposition velocities describes the range of uptake rates measured for many tree species and forest ecosystems (Hanson and Lindberg, 1991; Rondon and Granat, 1994; Hereid and Monson, 2001; Teklemariam and Sparks, 2006; Pape et al., 2008; Chaparro-Suarez et al., 2011; Delaria et al., 2018). Model calculations also predict a similar trend on the lifetimes of $NO_x$, with

a maximum reduction in the $NO_x$ lifetime by ~4 hrs (>40%) compared with no deposition.

       The large effect that small changes in stomatal conductance can have on $NO_x$ lifetimes, concentrations, budget, and $O_3$ production makes it very important to accurately parameterize in atmospheric models. Most global scale chemical transport models parameterize stomatal conductance using the representation developed by Wesely (1989) (Jacob and Wofsy, 1990; Verbeke et al., 2015). These do no account for the effects of VPD, SWP, $CO_2$ mixing ratio, or other factors known to influence

stomatal conductance (Hardacre et al., 2015). We show that incorporating vapor pressure deficit and soil water potential—using the parameterization of Emberson et al. (2000)—has a substantial impact on predicted $NO_2$ deposition, with the percent of soil $NO_x$ removed within the canopy increasing from 18% to 30% in wet (low VPD and high SWP) conditions compared to dry conditions in the location of BEARPEX-2009. Under the Wesely model, where stomatal conductance is parameterized only with temperature and solar radiation, the predicted deposition velocity would be nearly identical between "wet" and "dry"

days and between the spring and fall in semi-arid regions (e.g. much of the western United States, the Mediterranean Basin, the west coast of South America, parts of northwest Africa, parts of western and southern Australia, and parts of South Africa). The dominant effect of stomatal opening on $NO_2$ deposition causes an important time of day and seasonal behaviour that should be extensively explored with observations of $NO_x$ fluxes and concurrent models to confirm the role of deposition in a wider range of environs and more thoroughly vet the conceptual model proposed here.

*Code availability.* The model reported in this paper has been deposited in GitHub, https://github.com/erd02011/NOxmodel_ACP2019.

*Data availability.* The data presented in this study from Blodgett Forest and the University of Michigan Biological field Station
have been published in earlier work by Min et al. (2014) and Geddes and Murphy (2014), respectively.

*Author contributions.* ERD built the model, preformed data analysis, and prepared all figures. ERD wrote the manuscript in consultation with RCC. RCC supervised the project.

5    *Competing interests.* The authors declare that they have no conflict of interest.

*Acknowledgements.* We wish to gratefully acknowledge financial support from the National Science Foundation (NSF, AGS-1352972). This study was supported by NOAA Climate Program Office's Atmospheric Chemistry, Carbon Cycle, and Climate program NA18OAR4310117. Additional support was provided by an NSF Graduate Research Fellowship to Erin R. Delaria.
10   We would also like to give a special thanks to Jennifer G. Murphy, University of Toronto and Jeffrey Geddes, Boston University for providing data from the UMBS field site; and J. Geddes for constructive comments that improved the manuscript.

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

**Tables**

**Table 1: Parameters used in the model for comparison to observations from UMBS and BEARPEX-2009**

| Parameter | Symbol | UMBS | BEARPEX |
|---|---|---|---|
| Canopy height | $h_{can}$ | [a]20 m | [b]10 m |
| Understory height | $h_{us}$ | [c]4 m | [b]2 m |
| Total leaf area index | LAI | [c]3.5m$^2$/m$^2$ | [b]5.1 m$^2$/m$^2$ |
| Radiation extinction coefficient | $k_{rad}$ | [a]0.4 | [a]0.4 |
| Diffusion timescale ratio | $\tau/T$ | [a]2 | [a]2 |
| Friction velocity | $u*$ | [a]61 cm s$^{-1}$ | [a]61 cm s$^{-1}$ |
| Maximum NO emission flux | eNO$_{max}$ | [c]0.7 ppt ms$^{-1}$ | [b]3 ppt ms$^{-1}$ |
| Minimum NO emission flux | eNO$_{miin}$ | [c]0.3 ppt ms$^{-1}$ | [b]1 ppt ms$^{-1}$ |
| VOC basal emission flux | $E_b$ | [d]5 ppb m s$^{-1}$ | [b]11 ppb m s$^{-1}$ |
| Integration interval | $\Delta t$ | 2 | 2 |
| OH + VOC rate constant (cm$^3$ molecules$^{-1}$ s$^{-1}$) | kOH | [e]9.8× 10$^{-11}$ | [e]8.7× 10$^{-11}$ |
| NO$_3$ + VOC rate constant (cm$^3$ molecules$^{-1}$ s$^{-1}$) | kNO$_3$ | [e]7.0× 10$^{-13}$ | [e]1.7× 10$^{-14}$ |
| Minimum daily temperature | | 15 °C | 17 °C |
| Maximum daily temperature | | 23 °C | 27 °C |
| Maximum daily relative humidity | | 85% | 65% |
| Minimum daily relative humidity | | 65% | 30% |
| Maximum daily soil water potential | | [f]-0.05 MPa | [g]-0.8 MPa |
| Minimum daily soil water potential | | [f]-0.25 MPa | [g]-1.0 MPa |

a. Geddes and Murphy, 2014.
b. Wolfe and Thornton, 2011.
c. Seok et al., 2013
d. estimated from Bryan et al., 2015.
e. See text, calculated assuming dominant VOC is MBO for Blodgett and isoprene for UMBS
f. Estimated from Matheny et al., 2015.
g. Taken from Ishikawa and Bledsoe (2000) and Stern et al. (2018)

**Table 2: Parameters used in the Emberson model for stomatal conductance**

|  | UMBS | reference | BEARPEX | reference |
|---|---|---|---|---|
| $g_{max}$ (cm s$^{-1}$) | 0.2 | Büker et al. 2012 | 0.3 | Altimir et al. 2003 |
| $f_{min}$ | 0.05 | Büker et al. 2012 | 0.03 | Büker et al. 2012 |
| $Light\_a$ | 0.001 | Büker et al. 2012 | 0.001 | This study |
| $T_{max}$ (°C) | 33 | Büker et al. 2012 | 35 | Altimir et al. 2003 |
| $T_{min}$ (°C) | 5 | Büker et al. 2012 | 5 | Altimir et al. 2003 |
| $T_{opt}$ (°C) | 16 | Büker et al. 2012 | 20 | Altimir et al. 2003 |
| $VPD_{min}$ (kPa) | 3.1 | Büker et al. 2012 | 4 | Ryan et al. 2000, Hubbard et al. 2001, Kolb and Stone 1999 |
| $VPD_{max}$ (kPa) | 1.1 | Büker et al. 2012 | 1.5 | Ryan et al. 2000, Hubbard et al. 2001, Kolb and Stone 1999 |
| $SWP_{max}$ (MPa) | -1.0 | Emberson et al. 2000 | -1.0 | Anderegg et al. 2017 |
| $SWP_{min}$ (MPa) | -1.9 | Emberson et al. 2000 | -2.0 | Anderegg et al. 2017 |

**Figures**

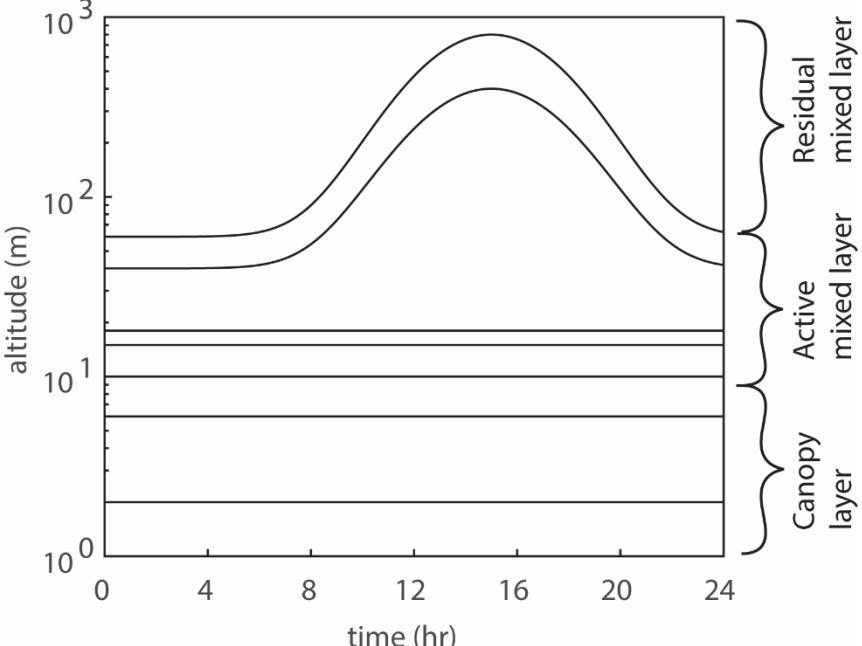

**Figure 1: Planetary boundary dynamics in the 1D multibox model. The model domain consists of three boxes in the canopy layer, four in the active mixed layer, and one in the residual mixed layer. The lower five boxes have fixed heights, while the sixth and seventh boxes evolve throughout the day, in the form of a Gaussian function.**

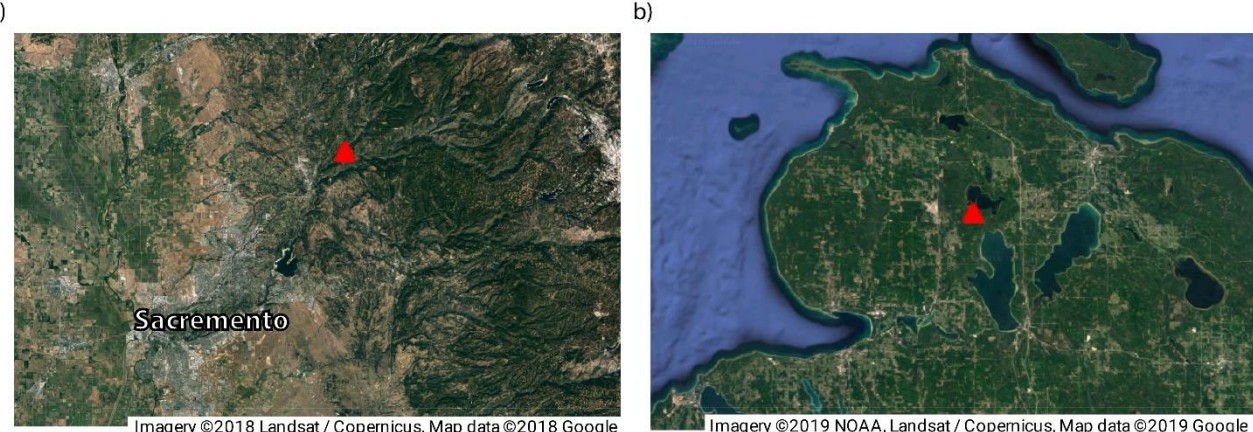

**Figure 2: Satellite images showing the locations of (a) the BEARPEX-2009 campaign and (b) the University of Michigan Biological Station (UMBS). Red triangles show the specific site locations. Measurements of chemical species and local meteorological variables from the two campaigns were used to validate our 1D canopy multibox model.**

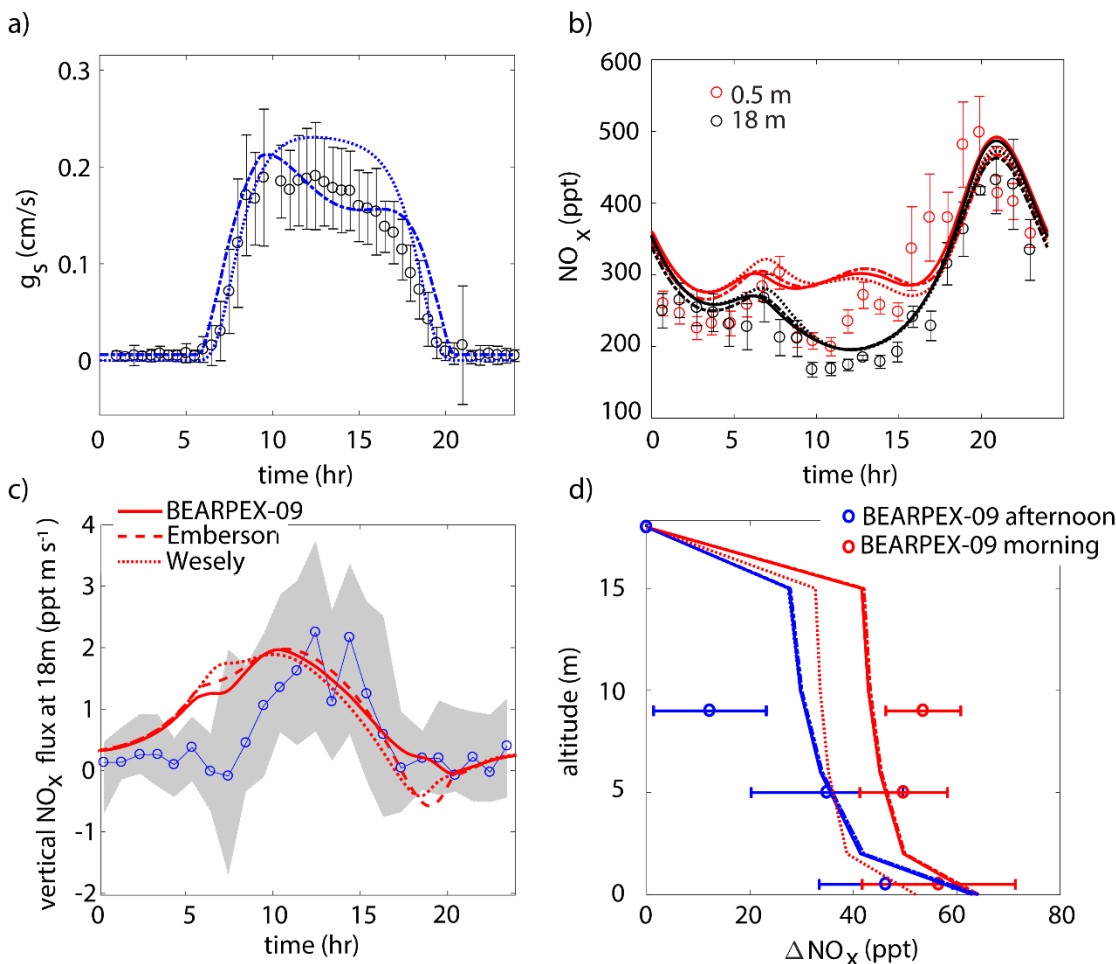

**Figure 3: Comparison of model results to BEARPEX-2009 hourly averaged observations of (a) stomatal conductances, (b) NOₓ mixing ratios at 18 m (black) and 0.5 m (red) and (c) vertical fluxes at 18 m. (d) Averaged observations of in-canopy NOₓ enhancements from 09:00–12:00 (blue) and 13:00–16:00 (red) compared with modeled NOₓ enhancements, defined as the difference between NOₓ below the canopy and NOₓ measured at 18 m. Observations from BEARPEX-2009 are from Min et al., (2014). In all panels solid lines, dotted lines, and dashed lines, represent results from our model with stomatal conductances parameterized using observed conductances, the Wesely model, and the Emberson model, respectively. Circles, error bars, and grey shaded regions represent observations, standard errors of the mean, and the interquartile range of data, respectively.**

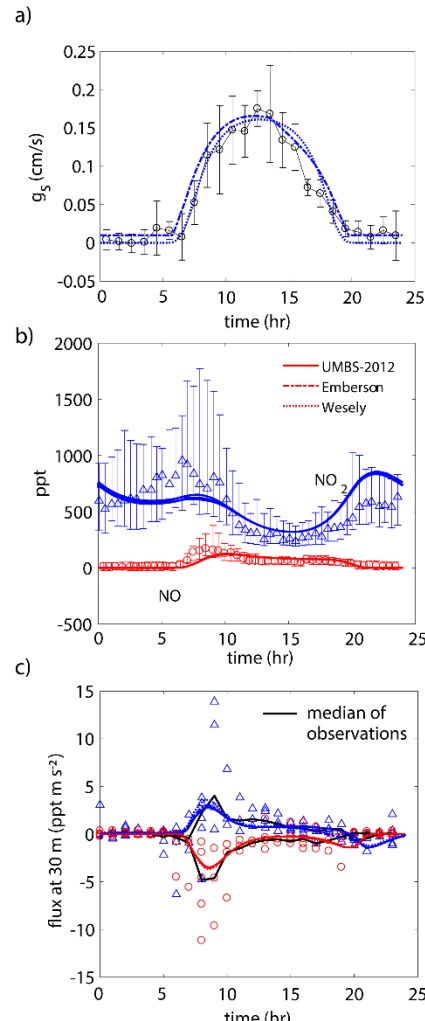

**Figure 4: Comparison of model results to (a) hourly averaged observed stomatal conductances, (b) NO and NO₂ mixing ratios at 30 m, and (c) median (black lines) and hourly-averaged NO and NO₂ vertical fluxes at 30 m observed during UMBS-2012 for August 8, 2012. In all panels solid lines, dotted lines, and dashed lines, represent results from our model with stomatal conductances parameterized using observed conductances, the Wesely model, and the Emberson model, respectively. Blue triangles and red circles represent NO₂ and NO observations, respectively. Error bars represent the interquartile range of data.**

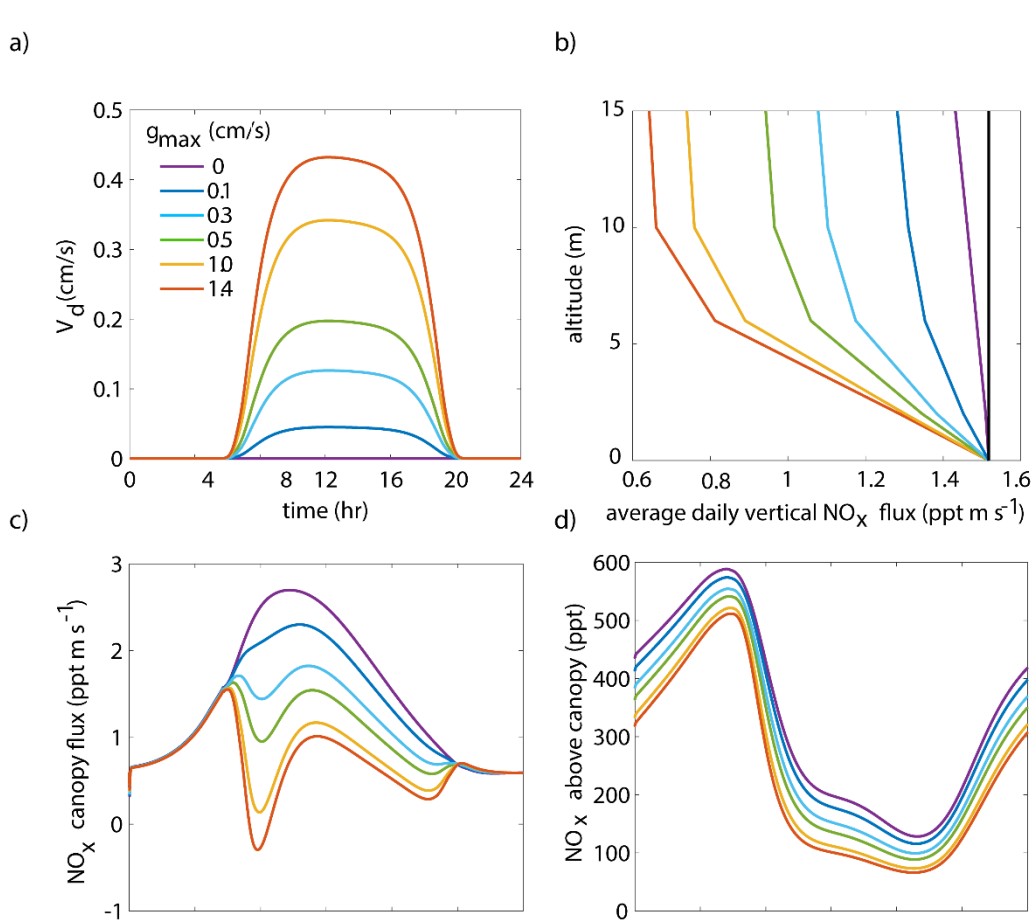

**Figure 5: Model results of (a) diurnal NO₂ deposition velocities, (b) average daily vertical fluxes of NOₓ and a conserved tracer (black line), (c) diurnal canopy fluxes at 10 m, and (d) diurnal above-canopy NOₓ mixing ratios at 15 m for different values of maximum stomatal conductance ($g_{max}$) using the Wesely scheme to calculate stomatal conductance.**

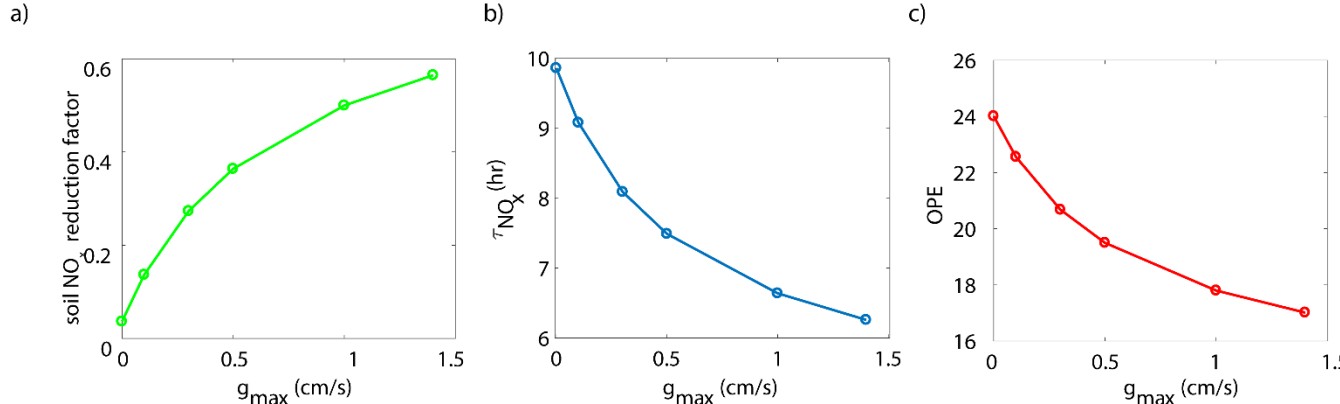

**Figure 6: Model-predicted dependence of (a) the fraction of soil emitted NOₓ removed in the canopy, (b) the average daily NOₓ lifetime ($\tau_{NO_x}$) in the planetary boundary layer, and (c) ozone production efficiency (OPE) on maximum stomatal conductance ($g_{max}$) using the Wesely scheme to calculate stomatal conductance.**

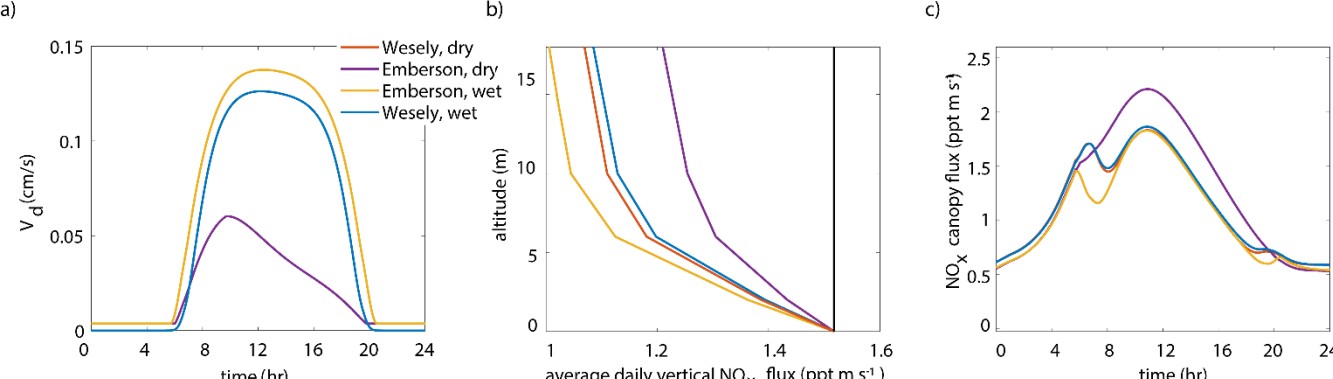

**Figure 7: Modeled results of (a) diurnal NO₂ deposition velocities, (b) average daily vertical fluxes compared to a conserved tracer (black line), and (c) diurnal canopy fluxes at 10 m for "wet" and "dry" scenarios using either the Wesely or Emberson models to calculate stomatal conductance.**

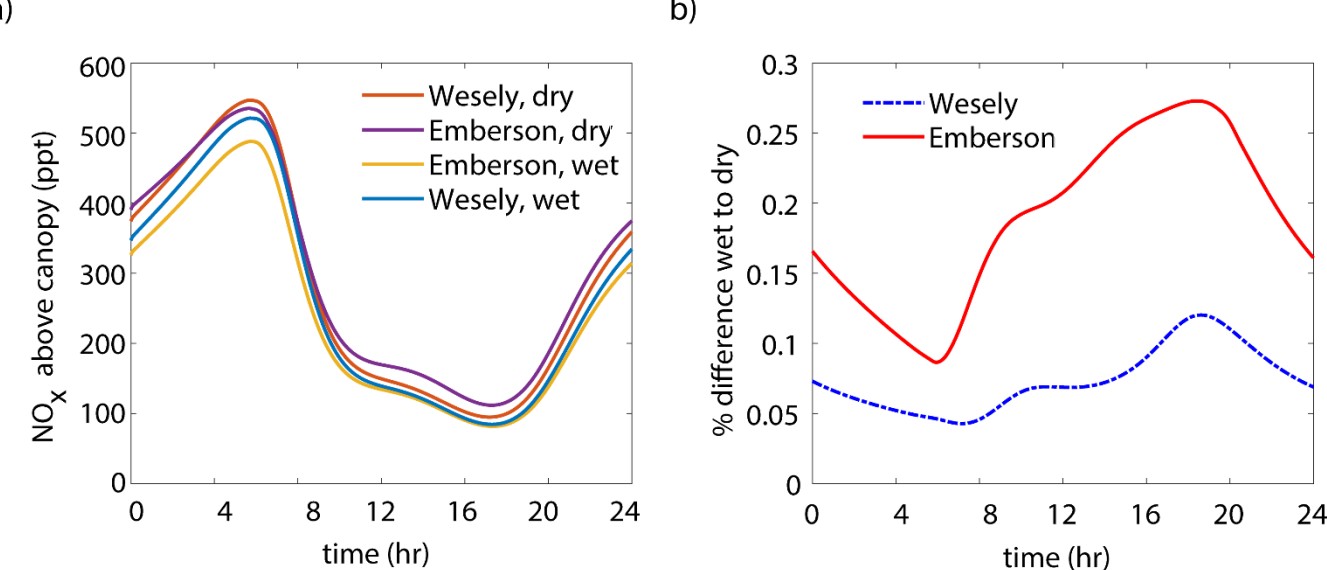

**Figure 8: (a) Modeled NO$_x$ mixing ratios above the canopy at 18 m for "wet" and "dry" scenarios using either the Wesely or Emberson models to calculate stomatal conductance. (b) Percent difference between NO$_x$ mixing ratios on "wet" and "dry" days using either the Wesely (blue dashed line) or Emberson (red solid line) parameterization of stomatal conductance.**

a)

b)

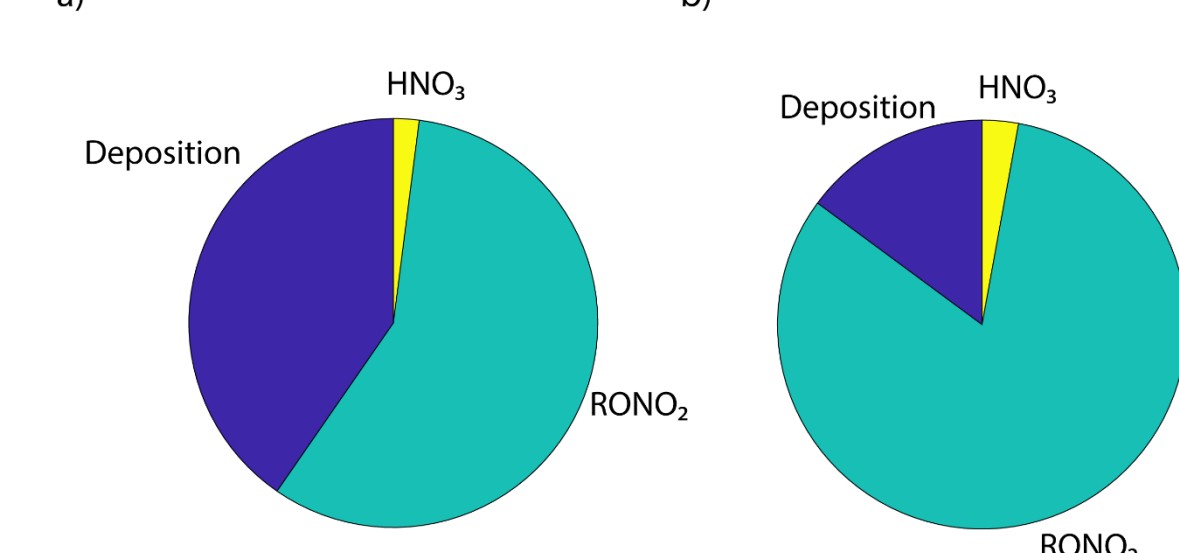

**Figure 9: Model prediction for the daytime average fraction of $NO_x$ removed by deposition, nitric acid formation, and alkyl nitrate formation using the Emberson parameterization of stomatal conductance for (a) "wet" and (b) "dry" conditions.**

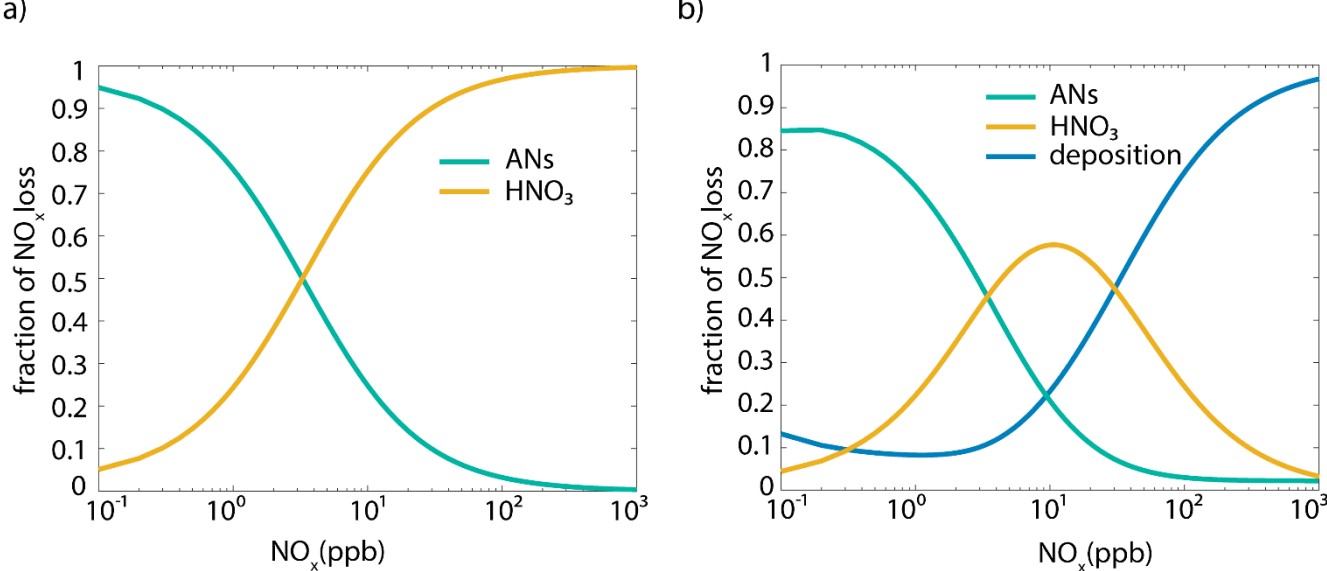

**Figure 10: Fraction of NO$_x$ loss to alkyl nitrate formation (green line), nitric acid formation (yellow line) with (a) no foliar uptake and (b) with foliar deposition (blue line) as a function of NO$_x$ mixing ratio predicted by the simplified single-box model.**