# Peer review of "A model-based analysis of foliar NOx deposition"

_Atmospheric Chemistry and Physics, 2019_

## Referee Comment (RC1) · Kirsti Ashworth (Referee) · 25 Aug 2019

The authors present a modeling study of NOx deposition to forest foliage using a 1-D column model consisting of 8 vertically-stacked boxes extending to the top of the PBL. Each box uses the continuity equation incorporating all production and loss processes occurring within and above a forest canopy. They apply this model at two contrasting forest ecosystems in the USA: a high-altitude Ponderosa pine forest in California and a northern deciduous mixed forest in Michigan from which they have a comprehensive suite of measurements from intensive field campaigns. They consider two commonly-used parameterisations of the stomatal deposition of ozone to model NOx deposition. They conduct a number of sensitivity studies to explore the processes that have the largest effect on modeled deposition velocities and quantify the subsequent impact on

fluxes and budgets of NOx, O3 and other key atmospheric gas-phase species within and above the canopy.

Loss rates of reactive gas-phase species, such as NOx, through deposition is a question of critical importance to our ability to model the complex interactions between NOx-HOx-Ox in the lower troposphere. While much effort has been put into improving parameterisations of O3 dry deposition, and particularly stomatal fluxes of O3, very little attention has been given to other gases. Yet the nitrogen cycle is fundamental to ecosystem functioning and health, the capacity of the terrestrial carbon sink and atmospheric composition (air quality and climate).

This study is highly timely and in my opinion excitingly novel and important. I commend the authors for their thorough and insightful approach to this global issue.

I have a couple of general concerns and a few specific comments that the authors should address prior to publication.

General:

1. While I appreciate that NO-NO2 cycling is rapid, non-linear and highly complex, I would like to authors to be explicit in precisely which molecule they are considering the dry deposition of. They rather inconsistently refer to deposition of NO2 and of NOx. Presumably they are assuming that NO deposition is negligible and hence deposition of NO2 can be used as a proxy of NOx deposition. If so, this should be explicitly stated early in the manuscript and a single term used from that point on.

2. The study purports to use two field sites, Blodgett Forest (BEARPEX campaign 2009) and University of Michigan Biological Station (UMBS campaign 2012). However, the authors almost exclusively focus their model description, parameterizations, sensitivity tests, results and discussions on Blodgett with scant details given to the results from UMBS other than to corroborate (or highlight differences) those from Blodgett. The authors should either reduce their analysis to a single site or, preferably, give similar attention to UMBS. The differences between model outcomes for the two sites is, to my mind, of real importance to enable the modeling and measurements communities to understand the processes that require further elucidation.

3. While the authors explicitly quantify the differences in NOx concentrations and fluxes between the two deposition schemes and between the perturbed parameter sensitivity tests, they do not similarly evaluate the relative performances against observations, relying instead on qualitative, descriptive differences. The results would be far stronger if this aspect of the model outcome were better explored and presented.

Specific:

Introduction

Throughout: All of the deposition models and studies presented here are specifically focused on the dry deposition of O3. The authors need to build a stronger argument that NO2 deposition should be assumed to follow the same process. In particular, in the case of O3, there still remain questions around the relative contributions of stomatal vs cuticular fluxes to the total leaf conductance. Most O3 deposition calculations assume that mesophyllic conductance is zero, is there evidence that this is the case for NO2.

p2, L19 (and elsewhere): VPD is a convenient proxy for leaf water potential as it can be calculated from routinely measured meteorological variables but it is often not a good metric to use under conditions of drought.

p2, L20: make clear that "season" and "seasonality" refers to plant phenology

p3, L4-5 (and elsewhere): Technically, the DO3SE model estimates stomatal conductance for use in deposition schemes to calculate deposition velocities and hence O3 fluxes.

p3, L12: Could the authors explicitly state some of these "other molecules"

p3, L15-17: YES!!! This should be emphasised!

**2 Model description**

p3, L21: A value of 100m for the PBL height during the peak growth season (summer) seems low, particularly for Blodgett. Under clear skies and high insolation I would expect to see values of 1500-2000m. Is their value based on observations at the two sites? If so, please provide references; if not please justify.

p3, 21: "Gaussian"

p3, L30: $\Delta h$ is surely the height / depth of the box. I assume that the model has a horizontal scale of 1m2 or 1cm2, but please clarify this.

p4, L1-19: This paragraph (which should really be split in two for BEARPEX and UMBS) is not a description of the model, rather the two field sites and should have a separate section.

p4, L7: I am surprised that UMBS was modelled here without a separate understory, see e.g. Bryan et al (2015) Atmos Environ.

p4, L20-21: Make clear here that this is simply following Beer's Law.

p4, L30: What are tau and TL in this context?

p4, L30 (and Table 1): Where is the value of u* taken from and why is it a constant value?

p5, L15: Please explain why the rate constants require adjustable parameters to make them site-specific. Are the authors assuming segregation? recycling?

p5, L21: Where are the basal emission rates taken from? Are they average values for deciduous and evergreen mid-latitude forests, site-specific, dominant-species specific?

p5, L22: Deposition should be described in a separate section. In fact, given it is the main focus of the study, it should be the first.

p5, L26-p6, L5: This is the Baldocchi parameterisation of total resistance. Why have

the authors not used the subsequent Gao et al (1993) update?

p6, L5-7: If all processes are correctly included and paratemerized there should be no need to use a compensation point; this is merely a formulation that is used when the production and loss terms are not fully represented in a model.

p6, L13: The authors have not defined SR

p6, L14: Eqn 12 is essentially the Jarvis (1976) parameterisation of stomatal conductance. It has been modified since, with additional adjustment factors. It forms the basis of the DO3SE model, but really the DO3SE model is about the damage and therefore incorporates an additional modifying factor fo3 to the Jarvis expression for gs.

p7, L6 & L8: VOC or BVOC?

p7, L16: Please expand on how fluxes are calculated within this model.

p7, L17: How is the PAN formation / NOx removal incorporated? It is not clear if or how these processes are included in the authors' considerations of chemical production and loss, lifetime calculations and OPE.

3 Sensitivity to parameterizations:

As previously noted, this section appears only to consider Blodgett Forest (unless all parameters were the same at both sites, which other parts of the manuscript suggest was not the case)

p7, L22-23: How were these values of total deposition velocity chosen?

p8, L10: Why have the authors chosen a value of 2 for tau/TL; Wolfe and Thornton (2011) used a value of 4 for this site when developing the CAFE model.

p8, L10: "resulting in a canopy residence time of 152s" at both sites? Or just Blodgett?

p8, L22: Please explain why Rb and specifically lw has a larger impact on species with high rates of leaf deposition.

p9, L14: I realise this is taken from a previous study but it is not clear why UMBS should be modeled using parameters for a European beech species when it is dominated by aspen.

p9, L19-p10, L4: Please quantify the model-obs fit rather than providing simply a qualitative overview.

p9, L25: Please explicitly state what is meant by NOx enhancement. I think it is the difference between in-canopy and above canopy concentrations. But these will differ between levels in the canopy and PBL

p10, L6: Wesely

p10, L26-28: How do these deposition velocities compare with observations? In L10, the authors state that values of 1.4, 0.77 and 1 are used in global models. Do the author have site-specific measurements on which they have based their choice of 0.3 and 1.4 as upper and lower bounds?

p11, L1: The authors are comparing 2 sites with a range of differences so I'm not sure they can claim "regional" differences. Surely it's more to do with different forest types, different soils, different meteorology, . . . Please could the authors be a little more specific.

P11, L2: I have a problem with the use of "wet" and "dry" in this context as deposition itself is referred to as wet or dry. Perhaps the authors can find an alternative way to describe wet and dry environments (I couldn't think of an obvious alternative I'm afraid)

p11, L4-7: Do these values of SWP and RH match long-term observations?

p11, L20-25: It would be good to see a more considered discussion of the results and the reasons (i.e the processes) behind the similarities and differences between the sites.

5 Discussion

p12, L6-7: Suggest the authors extend their view beyond the USA. Surely their findings are GLOBALLY applicable?

p12, L9: CLM includes a specific parameterization of stomatal conductance and is the land surface model for both regional and global models of chemistry-climate (see Lombardozzi et al, various). Models with a full land surface module already calculate stomatal conductance and plant physiology so have no need to incorporate either the Wesely or Emberson approaches for estimating gs.

p12, L21-22: Following on from the above point, this point about the relative simplicity of the Emberson approach should be made explicitly clear from the outset by the authors.

p13, L2: How is OPE defined? As molecule of O3 produced per "molecule" of NOx lost?

p13, L8: PBL

p13, L20-p14, L10: Parameterized for BEARPEX again?

p13, L26: Is PBL height fixed?

p14, L1 and L3: The plots of observed NOx concentrations for both sites suggest they are $\sim\leq$1ppb so why have the authors explored up to 100 ppb here?

6 Conclusions

p14, L25: missing closing parenthesis.

p14, L30-31: It's also imperative to accurately measure gs in a way that reflects differences between leaf-level and canopy-scale gs.

p14, L31-32: DO3SE is NOT a deposition model; it is a model of stomatal conductance that can be used in a deposition scheme so effectively it also uses the resistance in series approach.

[Figure]

p15, L14: Why is this important? What does this miss? Do we know that is wrong?

p15, L8: think GLOBAL!

p15, L8-9: Please could the authors be more specific in their recommendations? Precisely what do they mean by explore? More measurements? More modeling? And specifically of what, when and where?

Figures and Tables

p24, Fig 1: I am surprised that the authors have chosen only to vary PBL for the top two layers in the active mixed layer. I would expect the lower 2 of these layers to similarly evolve over the course of the day but with lower amplitude.

p24, Fig 1: Right-hand labels on plot say "remnant" and caption says "residual". Would personally use the latter.

p25, Fig 2: Not sure that this figure (or Fig S9) add anything to the paper. The authors have given the lat-long coordinates for both sites so readers could look for them on a map and they do not refer more than in passing to the figure from the text.

p26, Fig 3(d): Clarify what is meant by NOx enhancement in this context.

p27, Fig 4: A panel showing a time series of NOx would be helpful for direct comparison between the two sites.

p32, Fig 9: PAN? Daylight hours or 24-hour average?

Overall, I feel this is work of global significance that I hope will start to focus the attention of the atmospheric chemistry community on the importance of deposition in general and the fate of reactive nitrogen species in particular.

---

## Referee Comment (RC2) · Anonymous Referee #2 · 6 Sep 2019

This manuscript considers the importance of properly representing stomatal control in models of NOx deposition. The model framework is kept fairly simple as needed for anything that would be useful to larger-scale regional to global models. Two sites representing a dry montane pine forest and a mesic deciduous forest are considered. The model demonstrates reasonable skill at representing stomatal conductance and the resulting fluxes and concentrations of NO and NO2. Because stomatal conductance is so important, NOx deposition is shown to have diel and seasonal patterns influenced by moisture availability, which is not typically accounted for in standard resistance models. Overall, this paper presents a path forward for reconciling the literature on NOx canopy uptake factors using a more mechanistic approach. The paper is well written and provides a good description of the model and listing of the parameterizations used and

their sources. My only minor concern is that the figure captions and legends tend to be very terse and readers have to work to hard to figure out what the different symbols and lines mean. It would help to use few more words in the caption so it is clear what the lines and symbols in each panel are. When the same scheme in several panels is the same, just say so.
* * *

---

## Referee Comment (RC3) · Anonymous Referee #3 · 2 Oct 2019

The authors present a simple 8 layer box model based on gradient diffusion to study NOx retention by forests. The model is tested against data from two contrasting forest environments in the USA. A pine forest and a mixed hardwood forest. The study investigates an important aspect of atmospheric N-cycling that is not well represented in many models. NO emissions from soils below dense/tall canopies are partly retained by the canopy due to NO2 deposition or organic-N formation. This reduces the effect of soil NO emissions on atmospheric chemistry (e.g O3 formation) and N-burden. Given the importance of this topic I am supportive of publication after considering the comments given below.

Major comments:

While I appreciate that the model should be kept simple to allow broad applicability, I

am missing the discussion about limitations of this approach put by these simplifications. Especially, the effect of stability and coupling and decoupling of the forest canopy by coherent structures (e.g. Thomas and Foken, 2007; Sörgel et al., 2011; Steiner et al., 2011). Both effects can change residence times of compounds inside the canopy dramatically if compared to a near neutral stratification and homogeneous turbulence (which are underlying assumptions in the model equations used) and therefore influences the chemistry and the portion of NOx being transported out of the canopy. Furthermore, the authors state that: "Our model is able to closely replicate canopy fluxes and above-canopy NOx daytime mixing ratios observed during two field campaigns, one in a western Sierra Nevada pine forest (BEARPEX-2009) and the other a northern Michigan mixed hardwood forest (UMBS-2012)."

This implies that the model chemistry and physics are right because the model is able to reproduce measurements, but according to the authors "Advection concentrations are set to fit with the observations during BEARPEX-2009 (Min et al., 2014) or UMBS-2012 (Geddes and Murphy, 2014; Seok et al., 2013) and are used to maintain reasonable concentrations (Table S1)." The authors need to explain in more detail how the advection term was used to "maintain reasonable concentrations". In table S1 just one advection concertation is given for each site. Was this used as a constant background value throughout all simulations? How large is this advection term compared to other terms of the budget equation?

According to Fig. 3 and 4 the models have difficulties to reproduce fluxes correctly in the morning hours even if above canopy concentrations show reasonable agreement (below canopy concentrations are also not in agreement in the morning). For the BEARPEX campaign, measured fluxes are around zero until 7:30 while modelled fluxes increase from midnight onwards. This points to the influence of some of the processes not considered in the model (e.g stability and coherent exchange).

To be able to judge the abilities of the model to adequately reproduce the exchange the physical basis of the exchange parametrizations needs to be clearer. The authors

refer mostly to Wolfe and Thronton 2011. Unfortunately, the basic concepts underlying the derivation of the model equations are not discussed in detail in this work. What I could reconstruct is:

A) a relationship for in canopy u_star on above canopy u_star is given and the work of Yi et al. 2008 is cited as a source. Finnigan et al. 2015 showed that some of the assumptions of the work of Yi et al. 2008 are error prone. The relationship given might still be applicable but needs additional justification.

B) from the u_star profile a $\sigma$w profile is constructed by the relation given by Raupach et al., 1989 that $\sigma$w= 0.125 u_star.

C) Diffusivities are calculated finally by including a "near field correction factor (r) " that Makar et al. 1999 introduced to the far field diffusion equation given by Raupach (1989) to account for near field effects and to calculate the dispersion in a eulerian framework and not as originally used by Raupach an lagrangian. According to Wolfe and Thronton 2011 $\tau$/Tl is used to "tune" r, with typical values of $\tau$/Tl 1-4. According to Makar et al., 1999 r changes from about $\sim$0 to 0,99 in that range of $\tau$/Tl. Therefore approaching the far field (i.e. diffusive) regime. The canopies considered in this manuscript are comparably low and open. The canopy height is about 20 m and 10 m and the LAI is 3.5 m2m-2 and 3.2 m2m-2 (overstory)+ 1.9 m2m-2 (understory), which might explain the short residence time (< 3 min) in the model and may suffice the choice $\tau$/Tl >= 2 (domination of far field diffusion). However, this may not be easily generalized to taller and denser canopies. Therefore, I am puzzled that the authors cite ( on P8) the work of Jacob and Wofsy 1990 as to be "consistent" with the derived residence time < 3 min in this work as JW give the flushing time of the canopy (0-40 m) as 60 min during day and 300 min during night time!

As the residence time is essential in determining the chemistry and the deposition inside the forest this point needs to be further elaborated.

Specific comments:

P6 L4: "and are dependent upon plant physiology." => They also depend on the physical and chemical properties of the compounds. P8 L31: Did the different canopy shapes change the residence times or was this kept constant? Are canopy structure and LAI independent from the residence time in the model? P9 second paragraph: Here again the question how much influence has the "advection correction" here?

Technical comments:

P3 L21: "below the boundary layer" => replace by either "within the pbl" or "below pbl top" P8 L31: "is was" => is Fig.3 and 4: Please use same spacing of time axis for all panels. Makes it easier to compare. Figure 3d): which time intervals are used for "morning" and "afternoon"? Figure 4b): Move NO2 label in graph as the subscript 2 is hidden within the data points.

References:

Finnigan, J., Harman, I., Ross, A. and Belcher, S.: First-order turbulence closure for modelling complex canopy flows, Q. J. R. Meteorol. Soc., doi:10.1002/qj.2577, 2015.

Jacob D.J. and Wofsy S.C.: Budgets of reactive nitrogen, hydrocarbons, and ozone over the Amazon forest during the wet season. Journal of Geophysical Research, 95(D10),16737, https://doi.org/10.1029/JD095iD10p16737,1990.

Raupach, M. R.: A practical Lagrangian method for relating scalar concentrations to source distributions in vegetation canopies, Q. J. Roy. Meteor. Soc., 115, 609–632, 1989.

Sörgel, M., Trebs, I., Serafimovich, A., Moravek, A., Held, A., and Zetzsch, C.: Simultaneous HONO measurements in and above a forest canopy: influence of turbulent exchange on mixing ratio differences, Atmos. Chem. Phys., 11, 841–855, doi:10.5194/acp-11-841-2011, 2011.

Steiner, A. L., Pressley, S. N., Botros, A., Jones, E., Chung, S. H., and Edburg, S. L.: Analysis of coherent structures and atmosphere-canopy coupling strength during the

[Figure]

CABINEX field campaign, Atmos. Chem. Phys., 11, 11921–11936, doi:10.5194/acp-11-11921-2011, 2011.

Thomas, C. and Foken, T.: Flux contribution of coherent structures and its implications for the exchange of energy and matter in a tall spruce canopy, Bound.-Lay. Meteorol., 123, 317–337, 2007.

Wolfe, G. M. and Thornton, J. A.: The Chemistry of Atmosphere-Forest Exchange (CAFE) Model – Part 1: Model description and characterization, Atmos. Chem. Phys., 11, 77-101, https://doi.org/10.5194/acp-11-77-2011, 2011.

Yi, C.: Momentum Transfer within Canopies, J. Appl. Meteorol. Clim. 47, 262–275, doi:10.1175/2007JAMC1667.1, 2008.
* * *

---

## Author Comment (AC1) · 23 Oct 2019

We are very grateful for the constructive comments and valuable suggestions offered by the three reviewers. The reviewers' comments appear in **bold** followed by our responses to each comment in *italics*. Line numbers in our responses refer to the edited manuscript. Please see attached supplement for revisions referred to.

**Reviewer 1**

**General:**
**1. While I appreciate that NO-NO2 cycling is rapid, non-linear and highly**

[Figure]

**complex, I would like to authors to be explicit in precisely which molecule they are considering the dry deposition of. They rather inconsistently refer to deposition of NO2 and of NOx. Presumably they are assuming that NO deposition is negligible and hence deposition of NO2 can be used as a proxy of NOx deposition. If so, this should be explicitly stated early in the manuscript and a single term used from that point on.**

*We have gone through the manuscript and have corrected mention of "NOx deposition" to explicitly refer to "NO2 deposition". We also do use NO2 as a proxy of NOx deposition. A statement was added to P3, L17 clarifying that we consider NO deposition to be negligible.*

**2. The study purports to use two field sites, Blodgett Forest (BEARPEX campaign 2009) and University of Michigan Biological Station (UMBS campaign 2012). However, the authors almost exclusively focus their model description, parameterizations, sensitivity tests, results and discussions on Blodgett with scant details given to the results from UMBS other than to corroborate (or highlight differences) those from Blodgett. The authors should either reduce their analysis to a single site or, preferably, give similar attention to UMBS. The differences between model outcomes for the two sites is, to my mind, of real importance to enable the modelling and measurements communities to understand the processes that require further elucidation.**

*We have considered the reviewers comment. We believe similar giving similar attention to UMBS would distract from our focus on the conceptual conclusions of this paper. The purpose of including the UMBS data is to further corroborate the ideas in the model and to demonstrate that the model is applicable to multiple sites and not simply tuned for Blodgett Forest observations. The overall purpose of this manuscript is not*

*to argue that observations should exactly match our model predictions, but rather to illustrate trends and key ideas that field observations and modelling studies should pay further attention to in future observational and modelling research. We added to P11, L30: "Similar trends (not shown) were also observed using parameters for UMBS."*

*We also chose to focus our attention on Blodgett Forest for comparing the Wesely and Emberson models because this is a region subject to frequent dry conditions in the summer and fall, and view this site as an example of a region where our findings may be of particular importance.*

**3. While the authors explicitly quantify the differences in NOx concentrations and fluxes between the two deposition schemes and between the perturbed parameter sensitivity tests, they do not similarly evaluate the relative performances against observations, relying instead on qualitative, descriptive differences. The results would be far stronger if this aspect of the model outcome were better explored and presented.**

*We agree with the reviewer that it will be important in the future to directly and quantitatively compare models to observations. However, at this point in time, we believe clarifying key variables that govern NOx fluxes is an important advance even without such a quantitative comparison. Moreover, we are not aware of observations for a location during both dry and wet conditions. We call for more long-term observations of stomatal behaviour and dry deposition processes over a variety of meteorological conditions.*

**Specific:**
**Introduction**

**Throughout: All of the deposition models and studies presented here are specifically focused on the dry deposition of O3. The authors need to build a stronger argument that NO2 deposition should be assumed to follow the same process. In particular, in the case of O3, there still remain questions around the relative contributions of stomatal vs cuticular fluxes to the total leaf conductance. Most O3 deposition calculations assume that mesophyllic conductance is zero, is there evidence that this is the case for NO2.**

*We added a statement to P2, L27-29 with citations arguing NO2 deposition is also controlled by stomatal opening. Mesophyllic resistance in models is indeed assumed to be comparatively small. However, this is a question we are actively researching with laboratory chamber measurements. This will be followed up in a future publication currently in preparation.*

**p2, L19 (and elsewhere): VPD is a convenient proxy for leaf water potential as it can be calculated from routinely measured meteorological variables but it is often not a good metric to use under conditions of drought.**

*We agree with the reviewer comment. However, the focus that we take on VPD is indeed because it is a convenient proxy that we believe is practical. Consideration of VPD is a substantial improvement over current CTMs that do not include such a parameterization. We note that this does not completely tell the whole picture, which we discuss later P13, L19-31.*

**p2, L20: make clear that "season" and "seasonality" refers to plant phenology**

*"season" was changed to "seasonality of leaf phenology".*

**p3, L4-5 (and elsewhere): Technically, the DO3SE model estimates stomatal conductance for use in deposition schemes to calculate deposition velocities and hence O3 fluxes.**

*Line was changed to "...estimating stomatal conductance to predict ozone deposition velocities,...", now P3, L5-6.*

**p3, L12: Could the authors explicitly state some of these "other molecules"**

*P3, L19-20 now reads: "... other molecules such as NO2, NO, H2O2, HNO3, hydroxy nitrates, alkyl nitrates, peroxyacyl nitrates, etc...."*

**p3, L15-17: YES!!! This should be emphasised!**

*We agree, but are unsure what more we could do to emphasize this point.*

**2 Model description**
**p3, L21: A value of 100m for the PBL height during the peak growth season (summer) seems low, particularly for Blodgett. Under clear skies and high insolation I would expect to see values of 1500-2000m. Is their value based on observations at the two sites? If so, please provide references; if not please justify.**

*References to Wolfe and Thornton, (2011) and Wolfe et al., (2011) were added to P3, L29.*

**p3, 21: "Gaussian"**

*fixed*

**p3, L30: △h is surely the height / depth of the box. I assume that the model has a horizontal scale of 1m2 or 1cm2, but please clarify this.**

*Each box layer is treated as well-mixed and homogenous.*

**p4, L1-19: This paragraph (which should really be split in two for BEARPEX and UMBS) is not a description of the model, rather the two field sites and should have a separate section.**

*The paragraphs describing the two sites were separated into two paragraphs and a separate section added (2.2, P5, L17–P6, L12).*

**p4, L7: I am surprised that UMBS was modelled here without a separate under-story, see e.g. Bryan et al (2015) Atmos Environ.**

*There is a separate understory. This has been clarified in P5, L28 and Table 1. Citation to Bryan et al., 2015 was also added.*

**p4, L20-21: Make clear here that this is simply following Beer's Law.**

*". . .following Beer's law:" was added to P6, L9.*

**p4, L30: What are tau and TL in this context?**

*Please see Wolfe and Thornton, (2011). We added an additional citation of this paper following P6, L21. A definition was also added to P6, L20 : ".. defined as the ratio of the "time since emission" of a theoretical diffusing plume ($\tau$) and the Lagrangian timescale (TL). . ."*

**p4, L30 (and Table 1): Where is the value of u\* taken from and why is it a constant value?**

*We used for u\* the average daytime value reported by Wolfe and Thornton, (2011). The range of u\* during the BEARPEX-2009 campaign was $\sim$ 0.1–0.8. We decided to use the daytime average as a constant value, as for the most part we restricted our analysis to daytime results. We ran a scenario with our model in which u\* above the canopy varied based on a sinusoidal fit to average diurnal observations at Blodgett Forest, and observed negligible changes to the canopy fluxes and above-canopy NOx mixing ratios. Based on this, and our sensitivity test to $\tau/T_L$, we decided to leave out this additional complication in our model so that it would be easily extendable to forests where observations of u\* are not readily available.*

**p5, L15: Please explain why the rate constants require adjustable parameters to make them site-specific. Are the authors assuming segregation? recycling?**

*To P7, L23 we added the statement: "kOH and kNO3 are effective values adjusted in the model based on site-specific VOC composition and observations of OH reactivity."*

**p5, L21: Where are the basal emission rates taken from? Are they average values for deciduous and evergreen mid-latitude forests, site-specific, dominant-species specific?**

*Citations of the emissions rates and other parameters were added as a table caption for Table 1.*

**p5, L22: Deposition should be described in a separate section. In fact, given it is the main focus of the study, it should be the first.**

*We have rearranged the manuscript so Deposition appears in its own section and first in the section 2.1.*

**p5, L26-p6, L5: This is the Baldocchi parameterisation of total resistance. Why have the authors not used the subsequent Gao et al (1993) update?**

*The Baldocchi parameterization of total resistance is used because our model has been built to scale up laboratory observations of leaf-level deposition to the canopy scale. A similar approach was taken for CAFE model development (Wolfe and Thornton, 2011), on which this simplified model was based. In our opinion, the Gao update adds complexity without changing the aspects that are key to the discussion here.*

**p6, L5-7: If all processes are correctly included and parameterized there should be no need to use a compensation point; this is merely a formulation that is**

**used when the production and loss terms are not fully represented in a model.**

*We changed the sentence (P4, L21-22) to say: "We do not allow for emission of NO or NO2 from leaves, consistent with recent laboratory observations that have observed negligible compensation points for these molecules (Chaparro-Suarez et al., 2011; Breuninger et al., 2013; Delaria et al., 2018)."*

**p6, L13: The authors have not defined SR**

*A definition of SR has been added P5, L2.*

**p6, L14: Eqn 12 is essentially the Jarvis (1976) parameterisation of stomatal conductance. It has been modified since, with additional adjustment factors. It forms the basis of the DO3SE model, but really the DO3SE model is about the damage and therefore incorporates an additional modifying factor fo3 to the Jarvis expression for gs.**

*The Emberson et al. (2000) paper we refer to does not include this fO3 term. We added a citation of Jarvis et al. (1976) to P4, L28.*

**p7, L6 L8: VOC or BVOC?**

*BVOC. This has been updated p 7, L22.*

**p7, L16: Please expand on how fluxes are calculated within this model.**

*Fluxes are calculated according to Eq. 14 (updated manuscript). We added a reference to Wolfe and Thornton (2011) P6, L15, as the same method of calculating fluxes was used here. Reference to Eq. 14 was also added to P8, L14.*

**p7, L17: How is the PAN formation / NOx removal incorporated? It is not clear if or how these processes are included in the authors' considerations of chemical production and loss, lifetime calculations and OPE.**

*As shown in the Romer et al. reference, during the day at high temperatures, PAN is in steady state with NOx and a constant PAN/NOx ratio occurs. PANs role in these circumstances is to sequester NOx in a different form. In this paper, we neglect the possibility of direct PAN deposition. Upon deposition of NO2, PAN dissociates maintaining the fixed PAN/NOx ratio set by the steady-state. At night, PAN is assumed to be a permanent sink of NOx and not available to return to the NOx pool when NO2 is removed by deposition.*

*We have removed this discussion of night time chemistry/deposition as it is not important to the conclusions of the paper.*

**3 Sensitivity to parameterizations:**
**As previously noted, this section appears only to consider Blodgett Forest (unless all parameters were the same at both sites, which other parts of the manuscript suggest was not the case)**

**p7, L22-23: How were these values of total deposition velocity chosen?**

*We edited P8, L21-23 to read: "...based on values of gmax and gmin chosen for Blodgett forest (discussed above) and typical values for deposition velocity observed for a variety of species in the laboratory (Teklemariam and Sparks, 2006; Chaparro Suarez et al., 2011, Breuninger et al., 2013, Delaria et al., 2018). "*

**p8, L10: Why have the authors chosen a value of 2 for tau/TL; Wolfe and Thornton (2011) used a value of 4 for this site when developing the CAFE model.**

*In our simplified model, a value of 2 resulted in the residence time in the canopy most similar to what was observed at Blodgett Forest. The simplified model gave a different residence time with a value of 4 than in the CAFE model.*

**p8, L10: "resulting in a canopy residence time of 152s" at both sites? Or just Blodgett?**

*"...for Blodgett Forest,..." has been added to P9, L11 for clarification. We have also added the applicable UMBS residence time.*

**p8, L22: Please explain why Rb and specifically lw has a larger impact on species with high rates of leaf deposition.**

*At higher deposition velocities, the stomatal resistance is lower and Rb makes a greater contribution to the total resistance. We expect small changes in Rb under these conditions to have a greater overall effect. We have added to P9, L23: "...where Rb makes a greater contribution to the total resistance."*

**p9, L14: I realise this is taken from a previous study but it is not clear why UMBS should be modeled using parameters for a European beech species when it is dominated by aspen.**

*We agree with the reviewer that parameters for aspen would have been more appropriate. However, there is no available data we are aware of for the specific tree species found at UMBS. As the site also contains American beech trees, and other hardwood deciduous tree species, a European beech species was chosen as a "best guess" for how trees at UMBS would behave. We realize this is not ideal, and call for more studies of stomatal regulation of North American trees. We note that the resulting predictions are in plausible agreement with observations and that the parameters used are distinct form those at Blodgett Forest, serving our purpose of showing that the model parameters we identify as important are flexible enough to represent different ecosystems.*

**p9, L19-p10, L4: Please quantify the model-obs fit rather than providing simply a qualitative overview.**

*We added references to figures 3 and 4 where appropriate, as well as parentheticals describing quantitative differences to P10, L18-P11, L6.*

**p9, L25: Please explicitly state what is meant by NOx enhancement. I think it is the difference between in-canopy and above canopy concentrations. But these will differ between levels in the canopy and PBL**

*This has been clarified in P10, L16: ". . . ,relative to above-canopy mixing ratios, . . .". A definition has also been added to the caption for Figure 3.*

**p10, L6: Wesely**

*Fixed.*

**p10, L26-28: How do these deposition velocities compare with observations? In L10, the authors state that values of 1.4, 0.77 and 1 are used in global models. Do the author have site-specific measurements on which they have based their choice of 0.3 and 1.4 as upper and lower bounds?**

*An upper bound of 1.4 was chosen from the upper bound of the global model listed above. Our lower-bound estimate was 0.1 cm s-1, but we believe 0.3 cm s-1 is a more reasonable lower bound estimate based on chamber studies we have recently conducted. Quantitative data for 0.1 cm s-1 was added P11, L29-30 for consistency.*

**p11, L1: The authors are comparing 2 sites with a range of differences so I'm not sure they can claim "regional" differences. Surely it's more to do with different forest types, different soils, different meteorology, . . . Please could the authors be a little more specific.**

*We have made edits for accuracy on P12, L 2-3. The manuscript now reads: "The relative importance of including parameterizations of VPD and SWP in the calculation of stomatal conductance and overall deposition velocity is expected to be regionally variable, along with regional variations in dominant tree species, soil types, and meteorology."*

**P11, L2: I have a problem with the use of "wet" and "dry" in this context as deposition itself is referred to as wet or dry. Perhaps the authors can find an alternative way to describe wet and dry environments (I couldn't think of an obvious alternative I'm afraid)**

*We have considered the reviewers point and understand how referring to conditions as "wet" and "dry" is less than ideal. However, we also were unable to come up with a more appropriate way of referring to these conditions.*

**p11, L4-7: Do these values of SWP and RH match long-term observations?**

*Citations have been added to P12, L9-12 for our choices of "wet" and "dry" conditions.*

**p11, L20-25: It would be good to see a more considered discussion of the results and the reasons (i.e the processes) behind the similarities and differences between the sites.**

*The current discussion serves our purpose of showing that the model is plausibly related to a second location. More detailed analysis of similarities and differences strikes us as more appropriate when more extensive observations of NOx fluxes are available at a location.*

**p12, L6-7: Suggest the authors extend their view beyond the USA. Surely their findings are GLOBALLY applicable?**

*We considered the reviewer's suggestion, but we decided to leave as-is. We do not*

*feel that giving an example of a region with frequent droughts in the US implies our finding will not be applicable globally. Our intention was to give one such example of a type of environment that our findings may be important for.*

**p12, L9: CLM includes a specific parameterization of stomatal conductance and is the land surface model for both regional and global models of chemistry-climate (see Lombardozzi et al, various). Models with a full land surface module already calculate stomatal conductance and plant physiology so have no need to incorporate either the Wesely or Emberson approaches for estimating gs.**

**p12, L21-22: Following on from the above point, this point about the relative simplicity of the Emberson approach should be made explicitly clear from the outset by the authors.**

*We have added a line to the introduction to highlight the simplicity of the Emberson model. "We consider here both the Wesely model and the similarly simplistic approach of Emberson et al. (2000) that incorporates effects of VPD and SWP." We have also added a reference to the CLM P13, L24-26.*

**p13, L2: How is OPE defined? As molecule of O3 produced per "molecule" of NOx lost?**

*This definition is correct. Please see Eq. 26.*

**p13, L8: PBL**

*The suggested change has been made to P14, L18.*

**p13, L20-p14, L10: Parameterized for BEARPEX again?**

*All relevant parameterizations have been listed in this section. However, the values chosen for $\alpha$, VOC reactivity and PHOx were similar to conditions at BEARPEX-09. A clarification has been made in L17 of P14.*

**p13, L26: Is PBL height fixed?**

*Yes, this section describes a simple box-model that does not evolve in time.*

**p14, L1 and L3: The plots of observed NOx concentrations for both sites suggest they are $\sim\leq 1$ ppb so why have the authors explored up to 100 ppb here?**

*In our view, the purpose of a mechanistic model is to permit prediction outside the range of observations and to identify circumstances where a process is uniquely important. In this section, we explore the role of deposition in near-urban forests where NOx concentrations are significantly higher than the two forests we focus on as our test examples. We find that NOx loss via stomatally controlled deposition is the primary loss mechanism in cities. To our knowledge that idea is not described previously in the literature, at least not with a tool that has the potential for incorporation into quantitative modelling.*

**6 Conclusions**

**p14, L25: missing closing parenthesis.**

*Fixed.*

**p14, L30-31: It's also imperative to accurately measure gs in a way that reflects differences between leaf-level and canopy-scale gs.**

*We agree with the reviewer.*

**p14, L31-32: DO3SE is NOT a deposition model; it is a model of stomatal conductance that can be used in a deposition scheme so effectively it also uses the resistance in series approach.**

*The wording has been changed for accuracy on P16, L9.*

**p15, L1-4: Why is this important? What does this miss? Do we know that is wrong?**

*The text in the conclusions and paper points to several items that are important, including a mechanistic explanation for CRF, explicit modelling of stomatal opening, and recognition of NOx fluxes as a significant control over the NOx lifetime in a range of different circumstances. We do not believe it would help the reader for us to be repetitive on these points at this place in the text.*

**p15, L8: think GLOBAL!**

[Figure]

*Locations outside the US have been references P16, L17-18.*

**p15, L8-9: Please could the authors be more specific in their recommendations? Precisely what do they mean by explore? More measurements? More modeling? And specifically of what, when and where?**

*We have added a sentence to the end of the concluding paragraph:*

*". . . explored with observations of NOx fluxes and concurrent models to confirm the role of deposition in a wider range of environs and more thoroughly vet the conceptual model proposed here."*

**Figures and Tables**
**p24, Fig 1: I am surprised that the authors have chosen only to vary PBL for the top two layers in the active mixed layer. I would expect the lower 2 of these layers to similarly evolve over the course of the day but with lower amplitude.**

*We do not believe this additional complication would change the general themes presented here, although they would certainly change things in detail.*

**p24, Fig 1: Right-hand labels on plot say "remnant" and caption says "residual". Would personally use the latter.**

*Fixed.*

**p25, Fig 2: Not sure that this figure (or Fig S9) add anything to the paper. The**

**authors have given the lat-long coordinates for both sites so readers could look for them on a map and they do not refer more than in passing to the figure from the text.**

*It is our impression that other readers may appreciate having the maps. Particularly for observing the relative proximity to urban centers.*

**p26, Fig 3(d): Clarify what is meant by NOx enhancement in this context.**

*This has been clarified in the figure caption. NOx enhancement is defined as NOx at each height – NOx above the canopy.*

**p27, Fig 4: A panel showing a time series of NOx would be helpful for direct comparison between the two sites.**

*We prefer to only show the diurnal average and variance.*

**p32, Fig 9: PAN? Daylight hours or 24-hour average?**

*This figure shows an average of daylight hours. This has been clarified in the figure caption. PAN is included in NOx, as it is in steady-state with NOx during the day (Romer et al., 2016).*

**Reviewer 2**

*We have added additional legends to some of the figures. We have also gone through*

[Figure]

*our figure captions and tried to be as clear as possible about symbol meanings and more detailed in our descriptions where applicable.*

**Reviewer 3**

*We thank reviewer 3 for pointing out some of the complexities in representing canopy exchange. Here we have focused on a fairly simple representation because a model of this complexity is comparable to those utilized in regional or global models. We intend to focus less on the quantitative agreement and emphasize the key conceptual advances. We argue that to correctly represent the degree of complexity in atmosphere-biosphere interactions the new ideas we present are needed. With these ideas alone, we are able to reach some significant insight—especially that CRF's are not necessary. We do not intend to suggest that the ideas we present alone are adequate to describe canopy scale mixing. The parameterization used here is designed to simulate conditions in two forests. In response to reviewer 3, we have added the following text P6, L25-32.*

> *Our model is a simple parameterization of turbulent processes and as such will only capture mean vertical diffusion. Other work (Collineau and Brunet, 1993a; Raupach et al., 1996; Brunet and Irvine, 2000; Thomas and Foken, 2007; Sörgel et al., 2011; Steiner et al., 2011) has shown that "near-field" effects of individual canopy elements and coherent turbulent structures can play an important role in canopy exchange. These more intricate processes are not captured explicitly by our simple model. Previous work (Gao et al., 1993; Makar et al., 1999; Stroud et al., 2005; Wolfe et al., 2011) have also utilized fairly simple representations of canopy exchange in local and regional models As such, K-theory is likely sufficient to represent average vertical diffusion for the purposes of our study.*

*In response to the concerns presented by Reviewer 3, on page C3, we have added a more detailed description of the representation of mixing that we use in our model, along with specific citations of the works cited by Wolf and Thornton (2011) and Reviewer 3. We have added the following to the text P6, L20-24:*

*The details of the parameterization of turbulent diffusion fluxes is documented elsewhere (Wolfe and Thornton, 2011) and based on the works of Raupach (1989) and Makar et al. (1999). The height dependent friction velocity (u(z)\*) is attenuated from the above-canopy u\* according to Yi et al. (2008). Although Finnigan et al. (2015) identified flaws in this treatment, we believe it is sufficient for our focus on illustrating generalizable qualitative trends.*

*The following statement was added to P12, L30-33:*

*We recognize that the multibox model presented in this work is a simplified representation of physical processes, and as such is not likely to (and is not intended to) provide quantitative exactitude for the trends described above. However, we argue for the necessity of incorporating these conceptual advances for accurately representing canopy processes and predicting their effect on the NOx cycle.*

**Specific comments:**
**P6 L4: "and are dependent upon plant physiology." => They also depend on the physical and chemical properties of the compounds.**

*On page 4, L19-20 (originally P6, L4), we have included the statement: "Rleaf is dependent upon plant physiology and the chemical and physical properties of the*

*deposition compounds".*

**P8 L31: Did the different canopy shapes change the residence times or was this kept constant? Are canopy structure and LAI independent from the residence time in the model?**

*The different canopy shapes did change the residence time. The residence time for UMBS was added to P9, L11.*

**P9 second paragraph:**
**Here again the question how much influence has the "advection correction" here?**

*Specifics for how advection was treated in the model was added to P7, L10-11 and P11, L9.*

**Technical comments:**
**P3 L21: "below the boundary layer" => replace by either "within the pbl" or "below pbl top".**

*We changed this to "...within the planetary boundary layer (PBL)".*

**P8 L31: "is was" => is**

*Fixed*

**Fig.3 and 4: Please use same spacing of time axis for all panels. Makes it easier to compare.**

*Fixed*

**Figure 3d): which time intervals are used for "morning" and "afternoon"?**

*Interval definitions were added to the figure caption.*

**Figure 4b): Move NO2 label in graph as the subscript 2 is hidden within the data points.**

*Fixed*

Please also note the supplement to this comment:
https://www.atmos-chem-phys-discuss.net/acp-2019-538/acp-2019-538-AC1-supplement.pdf

---

## Editor Decision (ED1)

Dear authors, co-authors,

First of all, sorry for the delay in my decision on your paper "A model-based analysis
Of foliar NOx deposition" submitted for publication in ACP. I was extremely busy last 4-
5 weeks teaching full time and which didn't allow me to carefully evaluate once more again
your response to the provided three reviews/editors comment and the revised version of the
manuscript. Now that I have done so I see that you properly handled most of the comments
and revision. However, I cannot accept it yet for publication "as is" given that there were
still some minor issues that came up reading your ms once more again.

First of all, note that I am aware that some of these minor comments also might be
"appreciated" as an editor promoting his/her own work but mainly want to secure that the
different communities, the experimental-, air quality (AQ) and climate modelling
communities are properly informed about the state-of the art approaches. In the past I
noticed for example that there have been (AQ) studies being published that were ignorant
about chemistry-climate studies (also due to being published in different journals read by
the different communities?), such as the ones on model analysis of atmosphere-biosphere
exchange, as a function of stomatal conductance including soil moisture limitation. Your
study tackles an issue of relevance for all those communities and consequently support this
publication that hopefully further increases the awareness that currently applied approaches
to represent atmosphere-biosphere exchange in regional and global scale modelling studies
should be revised. You explicitly mention one of those reasons, to properly consider some
of the potentially relevant interactions and feedback mechanisms. But your study also
shows some more of the subtle features that must be further explored in both experimental
and modelling studies and hope that your study will reinforce those considerations.

Page 2, line 20., "$CO_2$"

Page 2: lines 23-25: Also to deal with my previous editors comment, you now mention the
study by Ganzeveld et al. "However, Ganzeveld et al. (2002a) implemented a multi-layer
column model in a global chemistry and general circulation model GCM-ECHAM
(European Centre Hamburg Model)..."
To avoid any potential misunderstanding seeing the reference to the paper on the single
column model study (Ganzeveld et al., 2002a), in the modified text on previous (large-
scale) studies on canopy NOx deposition, it could be interpreted that we included this
(atmospheric) column model in the global climate-chemistry system ECHAM. However,
we actually only included the multi-layer canopy exchange model system in ECHAM to
study the role of canopy interactions in global atmosphere-biosphere NOx exchange
(Ganzeveld et al., 2002b). The 2002a reference focused on an extensive evaluation of this
multi-layer canopy exchange model coupled to an atmospheric column model system, an
approach similar to the one presented in your study.

Section 2: I realized that there is one important omission that was previously not captured
by the reviewers nor by myself. In Figure 1 the vertical discretization of your column model
is shown but think it is important to also mention explicitly in Section 2 how many layers
actually represent the canopy and the overlaying atmosphere.

Page 11, line 34-35: Here you could also include the Ganzeveld et al. 2002$_a$ and $\underline{b}$ references
with these studies with the multi-layer canopy exchange modelling system both in a site-
scale as well as a global-scale set-up confirming the numbers regarding the effective
exchange of soil NOx reported by Jacob and Wofsy (tropical forest case) and Yienger and

Levy (global scale).

Some further minor comments that were triggered going through your reply and revision:

Reviewer #1

Comment: p3, L30: Δh is surely the height / depth of the box. I assume that the model has a horizontal scale of 1m2 or 1cm2, but please clarify this. 50

Response: "Each box layer is treated as well-mixed and homogenous"

Editor: This also addresses the issue I raised on providing some more information about the model set-up. Here I get the impression that you do not really address the comment: It is not so much about the vertical discretization but about for what of horizontal scale the model can be deemed being representative. Generally in such 1-D model approaches it is not so straightforward and being assumed that all the parameters are calculated for a $1m^2$ column but at the end the representative horizontal scale might be determined by the scale of the observations/data that are being included, e.g,, the scale of emission- or vegetation datasets used to constrain such model approaches.

Comment: p6, L5-7: If all processes are correctly included and parameterized there should be no need to use a compensation point; this is merely a formulation that is used when the production and loss terms are not fully represented in a model.

Response: We changed the sentence (P4, L29-30) to say: "We do not allow for emission of NO or NO2 from leaves, consistent with recent laboratory observations that have observed negligible compensation points for these molecules (Chaparro-Suarez et al., 2011; Breuninger et al., 2013; Delaria et al., 2018)."

Here it is worthwhile to mention that indeed more recent measurements indicate that there is no NO/NO2 compensation but older observations have suggested the presence of such a compensation point. This might also be due to measurement issues. But one other feature to bring in here, also given that you have evaluated your modelling system also for the UMBS site, the Seok et al. (2013) paper also reference in your study actually proposed the potential importance of an $NO_2$ compensation point for that site. This was based on a comparison of the observed and canopy exchange model simulated diurnal cycles in the $NO_x$ concentrations above and inside this forest canopy.

Comment: P11, L2: I have a problem with the use of "wet" and "dry" in this context as deposition itself is referred to as wet or dry. Perhaps the authors can find an alternative way to describe wet and dry environments (I couldn't think of an obvious alternative I'm afraid)

Response: We have considered the reviewers point and understand how referring to conditions as "wet" and "dry" is less than ideal. However, we also were unable to come up with a more appropriate way of referring to these conditions.

Editor, often a source of confusion, wet deposition versus dry deposition to wet surfaces. One suggestion: wet surface and dry surface deposition?

Comment: Figures and Tables
p24, Fig 1: I am surprised that the authors have chosen only to vary PBL for the top two 20 layers in the active mixed layer. I would expect the lower 2 of these layers to similarly

evolve over the course of the day but with lower amplitude.

Response: We do not believe this additional complication would change the general themes presented here, although they would certainly change things in detail.

Editor: although I tend to agree with you that most likely it will not largely effect your results, it would have been good to have included some results on a sensitivity analysis changing the vertical discretization of your modelling system as well as the representation of the PBL depth. If you have indeed done so in the development stage of your model it might be good to explicitly mention this.

---

## Author Response (AR3)

Page 2, line 20., "CO$_2$"

*Fixed*

Page 2: lines 23-25:

*This now reads: "However, Ganzeveld et al. (2002b) implemented a multi-layer column model in a global chemistry and general circulation model GCM-ECHAM (European Centre Hamburg Model) to study the role of canopy interactions in global atmosphere-biosphere NOx exchange and demonstrated the importance of considering interactions within the canopy, particularly in pristine forest sites."*

*We have made the above phrasing and citation changes. Please let of know if there are alternative specific phrasing changes we can make to more accurately characterize the cited works.*

Page 11, line 34-35:

*Citations to Ganzeveld et al. 2002a and 2002b has been added.*

Reviewer #1:

Comment: p3, L30:

*To clarify the spatial extent of the model, we have added the following statement to P4, L8-9.*

*"The model was designed to be representative of a homogenous forest environment with the aim of simulating observations at forest tower sites."*

*As a result the horizontal spatial extent can be taken to be on the order of ~10 km$^2$, the approximate footprint of the UMBS PROPHET site (Pressley et al., 2005).*

Pressley, S., Lamb, B., Westberg, H., Flaherty, J., Chen, J., and Vogel, C.: Long-term isoprene flux measurements above a northern hardwood forest, *J. Geophys. Res.*, 110, D07301, doi:10.1029/2004JD005523, 2005.

Comment: p6, L5-7:

*We have added a line to P10,L 9-11: "It should also be noted that this agreement was achieved without inclusion of an NO2 compensation point, whereas Seok et al. (2013) had proposed the importance of considering foliar NO2 emission at this location."*

Comment: P11, L2:

*We here we are not referring to wet and dry deposition, or dry deposition to wet and dry surfaces. Our use of "wet" and "dry" refers to the meteorology (eg high humidity and high soil moisture or low humidity and low soil moisture). The following statement was added to P12, L 11-12 for clarity:*

*". Here we use "dry" to refer to conditions of low humidity and low soil moisture and "wet"*
5 *to refer to conditions with high humidity and high soil moisture."*

Comment: Figures and Tables
p24, Fig 1:

[revised manuscript text omitted]

a)                                                      b)

[Figure]

Imagery ©2018 Landsat / Copernicus, Map data ©2018 Google          Imagery ©2019 NOAA, Landsat / Copernicus, Map data ©2019 Google

**Figure 2: Satellite images showing the locations of (a) the BEARPEX-2009 campaign and (b) the University of Michigan Biological Station (UMBS). Red triangles show the specific site locations. Measurements of chemical species and local meteorological variables from the two campaigns were used to validate our 1D canopy multibox model.**

[Figure]

**Figure 3: Comparison of model results to BEARPEX-2009 hourly averaged observations of (a) stomatal conductances, (b) $NO_x$ mixing ratios at 18 m (black) and 0.5 m (red) and (c) vertical fluxes at 18 m. (d) Averaged observations of in-canopy $NO_x$ enhancements from 09:00–12:00 (blue) and 13:00–16:00 (red) compared with modeled $NO_x$ enhancements, defined as the difference between $NO_x$ below the canopy and $NO_x$ measured at 18 m. Observations from BEARPEX-2009 are from Min et al., (2014). In all panels solid lines, dotted lines, and dashed lines, represent results from our model with stomatal conductances parameterized using observed conductances, the Wesely model, and the Emberson model, respectively. Circles, error bars, and grey shaded regions represent observations, standard errors of the mean, and the interquartile range of data, respectively.**

[Figure]

**Figure 4: Comparison of model results to (a) hourly averaged observed stomatal conductances, (b) NO and NO₂ mixing ratios at 30 m, and (c) median (black lines) and hourly-averaged NO and NO₂ vertical fluxes at 30 m observed during UMBS-2012 for August 8, 2012. In all panels solid lines, dotted lines, and dashed lines, represent results from our model with stomatal conductances parameterized using observed conductances, the Wesely model, and the Emberson model, respectively. Blue triangles and red circles represent NO₂ and NO observations, respectively. Error bars represent the interquartile range of data.**

[Figure]

a)

b)

c)

d)

**Figure 5: Model results of (a) diurnal NO₂ deposition velocities, (b) average daily vertical fluxes of NOₓ and a conserved tracer (black line), (c) diurnal canopy fluxes at 10 m, and (d) diurnal above-canopy NOₓ mixing ratios at 15 m for different values of maximum stomatal conductance ($g_{max}$) using the Wesely scheme to calculate stomatal conductance.**

a)

b)

c)

[Figure]

**Figure 6: Model-predicted dependence of (a) the fraction of soil emitted NO$_x$ removed in the canopy, (b) the average daily NO$_x$ lifetime ($\tau_{NO_x}$) in the planetary boundary layer, and (c) ozone production efficiency (OPE) on maximum stomatal conductance ($g_{max}$) using the Wesely scheme to calculate stomatal conductance.**

[Figure]

**Figure 7: Modeled results of (a) diurnal NO₂ deposition velocities, (b) average daily vertical fluxes compared to a conserved tracer (black line), and (c) diurnal canopy fluxes at 10 m for "wet" and "dry" scenarios using either the Wesely or Emberson models to calculate stomatal conductance.**

a)

b)

[Figure]

**Figure 8: (a) Modeled NOₓ mixing ratios above the canopy at 18 m for "wet" and "dry" scenarios using either the Wesely or Emberson models to calculate stomatal conductance. (b) Percent difference between NOₓ mixing ratios on "wet" and "dry" days using either the Wesely (blue dashed line) or Emberson (red solid line) parameterization of stomatal conductance.**

a)                b)

[Figure]

**Figure 9: Model prediction for the daytime average fraction of NO$_x$ removed by deposition, nitric acid formation, and alkyl nitrate formation using the Emberson parameterization of stomatal conductance for (a) "wet" and (b) "dry" conditions.**

[Figure]

**Figure 10: Fraction of NO$_x$ loss to alkyl nitrate formation (green line), nitric acid formation (yellow line) with (a) no foliar uptake and (b) with foliar deposition (blue line) as a function of NO$_x$ mixing ratio predicted by the simplified single-box model.**